# GALOIS: Boosting Deep Reinforcement Learning via Generalizable Logic Synthesis

**Yushi Cao**[2,*], **Zhiming Li**[2,*], **Tianpei Yang**[1,3,†], **Hao Zhang**[1], **Yan Zheng**[1,†]
**Yi Li**[2], **Jianye Hao**[1], **Yang Liu**[2]
[1]College of Intelligence and Computing, Tianjin university, Tianjin, China
[2]Nanyang Technological University, Singapore, [3]University of Alberta, Canada
{yushi002,zhiming001}@e.ntu.edu.sg
{tpyang,3018216216,yanzheng,jianye.hao}@tju.edu.cn
{yi_li,yangliu}@ntu.edu.sg

## Abstract

Despite achieving superior performance in human-level control problems, unlike humans, deep reinforcement learning (DRL) lacks high-order intelligence (e.g., logic deduction and reuse), thus it behaves ineffectively than humans regarding learning and generalization in complex problems. Previous works attempt to directly synthesize a white-box logic program as the DRL policy, manifesting logic-driven behaviors. However, most synthesis methods are built on imperative or declarative programming, and each has a distinct limitation, respectively. The former ignores the *cause-effect* logic during synthesis, resulting in low generalizability across tasks. The latter is strictly proof-based, thus failing to synthesize programs with complex hierarchical logic. In this paper, we combine the above two paradigms together and propose a novel **G**ener**a**lizable **Lo**g**i**c **S**ynthesis (**GALOIS**) framework to synthesize hierarchical and strict *cause-effect* logic programs. GALOIS leverages the program sketch and defines a new sketch-based hybrid program language for guiding the synthesis. Based on that, GALOIS proposes a sketch-based program synthesis method to automatically generate white-box programs with generalizable and interpretable cause-effect logic. Extensive evaluations on various decision-making tasks with complex logic demonstrate the superiority of GALOIS over mainstream baselines regarding the asymptotic performance, generalizability, and great knowledge reusability across different environments.

## 1 Introduction

Deep reinforcement learning (DRL) has achieved great breakthroughs in various domains like robotics control [27], video game [24], software testing [48, 51, 5], etc. Despite its sheer success, DRL models still perform less effective learning and generalization abilities than humans in solving long sequential decision-making problems, especially those requiring complex logic to solve [16, 40]. For example, a seemingly simple task for a robot arm to put an object into a drawer is hard to solve due to the complex intrinsic logic (e.g., open the drawer, pick the object, place the object, close the drawer) [33]. Additionally, DRL policies are also hard to interpret since the result-generating processes of the neural network remain opaque to humans due to its black-box nature [35, 28].

To mitigate the above challenges, researchers seek the programming language, making the best of both connectionism [30] and symbolism [43], to generate white-box programs as the policy to

*Equal contribution.
†Corresponding authors: Yan Zheng (yanzheng@tju.edu.cn) and Tianpei Yang (tpyang@tju.edu.cn).

36th Conference on Neural Information Processing Systems (NeurIPS 2022).

execute logic-driven and explainable behaviors for task-solving. Logic contains explainable task-solving knowledge that naturally can generalize across similar tasks. Therefore, attempts have been made to introduce human-defined prior logic into the DRL models [46]. Human-written logic programs are found to be an effective way to improve the learning performance and zero-shot generalization [40]. However, such a manner requires manually written logic programs beforehand for each new environment, motivating an urgent need for automatic program synthesis.

Existing program synthesis approaches can be categorized into two major paradigms: imperative and declarative programming [6, 36, 29], each has its distinct limitation. The imperative programming aims to synthesize multiple sub-programs, each has a different ability to solve the problem, and combine them sequentially as a whole program [44, 15, 17]. However, programs synthesized in such a way has limited generalizability and interpretability since the imperative programming only specify the *post-condition* (*effect*) while ignores the *pre-condition* (*cause*) of each sub-program, which is regarded as a flawed reflection of causation [8] that is prone to *aliasing*. In other words, the agent will arbitrarily follow the synthesized program sequentially without knowing why (i.e., *cause-effect* logic). For example, assume a task in Figure 1 that requires the agent to open the box, get the key, open the door, then reach the goal. The synthesized imperative program would contain sub-programs: `toggle_box()`; `get_key()`; `open_door()`; `reach_goal()`, each should be executed sequentially (the blue path). However, when applying such a program to another similar task with minor logical differences: the key is placed outside the box, meaning the agent does not need to open the box. The synthesized program becomes sub-optimal as the agent will always follow the program to open the box first. However, the optimal policy should directly head for the key and ignores the box (denoted as the orange path).

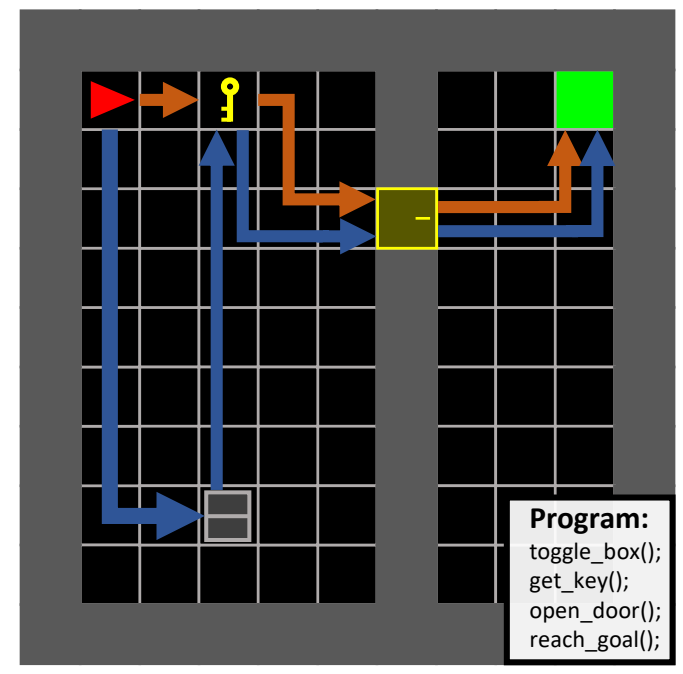

Figure 1: A motivating example.

On the other side, declarative programming aims to synthesize programs with explicit *cause-effect* logic [16, 10] in the form of first-order logic (FOL) [26], requiring the programs to be built on the proof system (i.e., verify the trigger condition given the facts, then decide which rule should be activated) [4]. However, due to the trait of FOL, programs synthesized in this way lack hierarchical logic and thus are ineffective in solving complex tasks [1].

To combine the advantages of both paradigms and synthesize program with hierarchical *cause-effect* logic, we propose a novel **G**ener**a**lizable **Lo**g**i**c **S**ynthesis (GALOIS) framework[3] for further boosting the learning ability, generalizability and interpretability of DRL. First, GALOIS introduces the concept of the program sketch [38] and defines a new hybrid sketch-based domain-specific language (DSL), including the syntax and semantic specifications, allowing synthesizing programs with hierarchical logic and strict *cause-effect* logic at the same time. Beyond that, GALOIS proposes a sketch-based program synthesis method extended from the differentiable inductive logic programming [12], constructing a general way to synthesize hierarchical logic program given the program-sketch. In this way, GALOIS can not only generate hierarchical programs with multiple sub-program synergistic cooperation for task-solving but also can achieve strict *cause-effect* logic with high interpretability and generalizability across tasks. Furthermore, the synthesized white-box program can be easily extended with expert knowledge or tuned by humans to efficiently adapt to different downstream tasks. Our contributions are threefold: (1) a new sketch-based hybrid program language is proposed for allowing hierarchical logic programs for the first time, (2) a general and automatic way is proposed to synthesize programs with generalizable *cause-effect* logic, (3) extensive evaluations on various complex tasks demonstrate the superiority of GALOIS over mainstream DRL and program synthesis baselines regarding the learning ability, generalizability, interpretability, and knowledge (logic) reusability across tasks.

[3]The implementation is available at: `https://sites.google.com/view/galois-drl`

## 2 Preliminary

### 2.1 Markov Decision Process

The sequential decision-making problem is commonly modeled as a Markov decision process (MDP), which is formulated as a 5-tuple $(S, A, R, P, \lambda)$, where $S$ is the state space, $A$ is the action space, $R : S \times A \rightarrow \mathbb{R}$ is the reward function, $P : S \times A \rightarrow S$ is the transition function, and $\lambda$ is the discount factor. The agent interacts with the environment following a policy $\pi(a_t|s_t)$ to collect experiences $\{(s_t, a_t, r_t)\}_{t=0}^{T}$, where $T$ is the terminal time step. The goal is to learn the optimal policy $\pi^*$ that maximizes the expected discounted return: $\pi^* = \arg\max_\pi \mathbb{E}_{a\sim\pi}[\sum_{t=0}^{T} \lambda^t r_t]$.

### 2.2 Inductive Logic Programming

Logic programming is a programming paradigm that requires programs to be written in a definite clause, which is of the form: $H :- A_1, ..., A_n$, where $H$ is the head atom and $A_1, ..., A_n, n \geq 0$ is called the body that denotes the conjunction of $n$ atoms, $:-$ denotes logical entailment: $H$ is true if $A_1 \wedge A_2 ... \wedge A_n$ is true. An atom is a function $\psi(\omega_1, ..., \omega_n)$, where $\psi$ is a $n$-ary predicate and $\omega_i, i \in [1, n]$ are terms. A predicate defined based on ground atoms without deductions is called an extensional predicate. Otherwise, it is called an intensional predicate. An atom whose terms are all instantiated by constants is called a ground atom. The ground atoms whose propositions are known in prior without entailment are called facts. Note that a set composed of all the concerning ground atoms is called a Herbrand base.

Inductive Logic Programming (ILP) [19] is a logic program synthesis model which synthesizes a logic program that satisfies the pre-defined specification. In the supervised learning setting, the specification is to synthesize a logic program $C$ such that $\forall \zeta, \lambda : F, C \models \zeta,\ F, C \not\models \lambda$, where $\zeta, \lambda$ denotes positive and negative samples, $F$ is the set of background facts given in prior; and for the reinforcement learning setting, the specification is to synthesize $C$ such that $C = \mathrm{argmax}_C\, R$, where $R$ is the average return of each episode. Specifically, ILP is conducted based on the valuation vector $\mathbf{e} \in \{0, 1\}^{|G|}$, $G$ denotes the Herbrand base of the ground atoms. Each scalar of $\mathbf{e}$ represents the true value of the corresponding ground atom. During each deduction step, $\mathbf{e}$ is recursively updated with the forward chaining mechanism, such that the auxiliary atoms and target atoms would be grounded.

## 3 Methodology

### 3.1 Motivation

As aforementioned, solving real decision-making problems, e.g., robot navigation and control [44, 34, 45], commonly requires complicated logic. As humans, we use the "divide-and-conquer" concept to dismantle problems into sub-problems and solve them separately. It is natural to think of generating a hierarchical logic program to solve complex problems. This intuition, however, has hardly been adopted in program synthesis since the strict *cause-effect* logic program is intrinsically non-trivial to generate, let alone the one with hierarchical logic [36, 29].

In this work, we propose a generalized logic synthesis (GALOIS) framework for synthesizing a white-box hierarchical logic program (as the policy) to execute logically interpretable behaviors in complex problems. Figure 2 shows the overview of GALOIS, comprised of two key components: ❶ a sketch-based DSL, and ❷ a sketch-based program synthesis method. It is noteworthy that GALOIS uses a white-box program as the policy to interact with the environment and collect data for policy optimization. Here, a new DSL is defined for creating hierarchical logic programs; and the sketch-based program synthesis method based on differentiable ILP is adopted for generating effective logic for the policy synthesis. In this way, GALOIS can synthesize white-box programs with generalizable logic more efficiently and automatically.

### 3.2 Sketch-based Program Language

To synthesize logic programs with both hierarchical logic and explicit *cause-effect* logic, we design a novel sketch-based DSL, namely $\mathcal{L}_{\text{hybrid}}$, absorbing both the advantages of imperative and declarative programming. Figure 3 shows the detail syntax and semantic specifications of $\mathcal{L}_{\text{hybrid}}$. It is

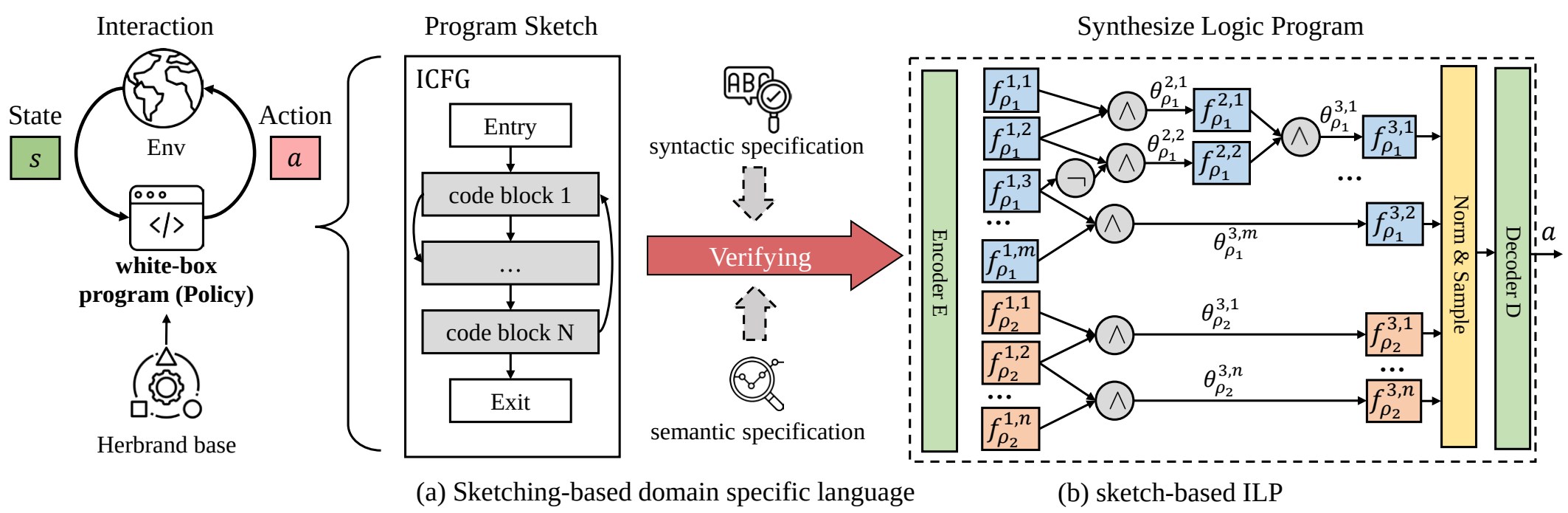

Figure 2: Overview of GALOIS, where the (a) sketch-based DSL defines what program can be synthesized, and (b) sketch-based ILP synthesizes programs with logic where $f_{\rho}^{\tau,\psi}$ represents predicate $\psi$ of hole function $\rho$ at inference step $\tau$ and $\theta_{\rho}^{\tau,\psi}$ is the corresponding weight.

Imperative:

$$e ::= n \mid f \mid \perp$$
$$c ::= x := e \mid c \,;\, c \mid \textbf{while } e \textbf{ do } c$$

Declarative:

$$f ::= R \mid f\ R \mid ??$$
$$R ::= A \text{ ``: -'' } A\text{-}list$$
$$A\text{-}list ::= A \mid A \wedge A\text{-}list$$

(a) syntax

$$\textbf{WHERE}\quad \mathcal{C}[\![??_{\text{WHERE}}]\!]\langle s, \Phi\rangle = \langle s[@ \mapsto \mathcal{C}[\![\mathcal{A}_{\text{WHERE}}(s) \models g]\!]s], \Phi\rangle$$

$$\textbf{HOW}\quad \mathcal{C}[\![??_{\text{HOW}}]\!]\langle s, \Phi\rangle = \langle s[\text{ pos } \mapsto \mathcal{C}[\![\mathcal{A}_{\text{HOW}}(@) \models d]\!]s], \Phi\rangle$$

$$\textbf{WHAT}\quad \mathcal{C}[\![??_{\text{WHAT}}]\!]\langle s, \Phi\rangle = \langle s[o \mapsto \mathcal{C}[\![\mathcal{A}_{\text{WHAT}}(s) \models a]\!]s], \Phi\rangle$$

(b) semantics

Figure 3: The (a) syntactic and (b) semantic specifications of DSL $\mathcal{L}_{\text{hybrid}}$.

noteworthy that $\mathcal{L}_{\text{hybrid}}$ ensures the synthesized program follow strict *cause-effect* logic. Beyond that, following $\mathcal{L}_{\text{hybrid}}$, we synthesize programs using program sketches, allowing generating hierarchical logic programs. In the following, we describe the formal syntactic and semantic specifications first and illustrate how the program sketch derives hierarchical logic programs.

**Syntactic Specification:** The formal syntactic specifications of $\mathcal{L}_{\text{hybrid}}$ are defined using elements from both the declarative and imperative language (shown in Figure 3(a)). Intuitively, the declarative language demands the synthesized program follow strict *cause-effect* logic, while the imperative language enables programs with hierarchical logic. In specific, imperative language elements are expression $e$ and command $c$. Term $e$ can be instantiated as constant $n$ or function call $f$, and $c$ can be assignment statement $x := e$, sequential execution $c; c$ or control flow (**while** loop). Declarative language elements are function $f$ and clause $R$. To expose *cause-effect* relations, we constrains functions to be implemented declaratively: $f ::= R \mid f\ R$, where $R$ represents logic clause in the form of $R ::= A$":-"$A\text{-}list$, where $A$ denotes atom and $A\text{-}list$ is the clause body.

It is noteworthy that, to implement the *program sketch*, we introduce a novel language element called *hole function*, denoted as $??$. This hole function denotes an unimplemented logic sub-program (i.e., code block in Figure 2) to be synthesized given the constraints specified by the program sketch and its corresponding semantic specification.

**Semantic Specification:** Having the syntactic specification, any syntactically valid program sketch can be derived. However, without semantic guidance (e.g., lack of task-related semantics), the synthesized program may lack sufficient hierarchical logic to efficiently solve tasks [52]. Hence, we propose leveraging the program sketch [38] and defining associated semantic specifications to guide the synthesis to generate hierarchical logic programs. In the following, we illustrate the details of the program sketch used in this work and its formal semantic specifications,

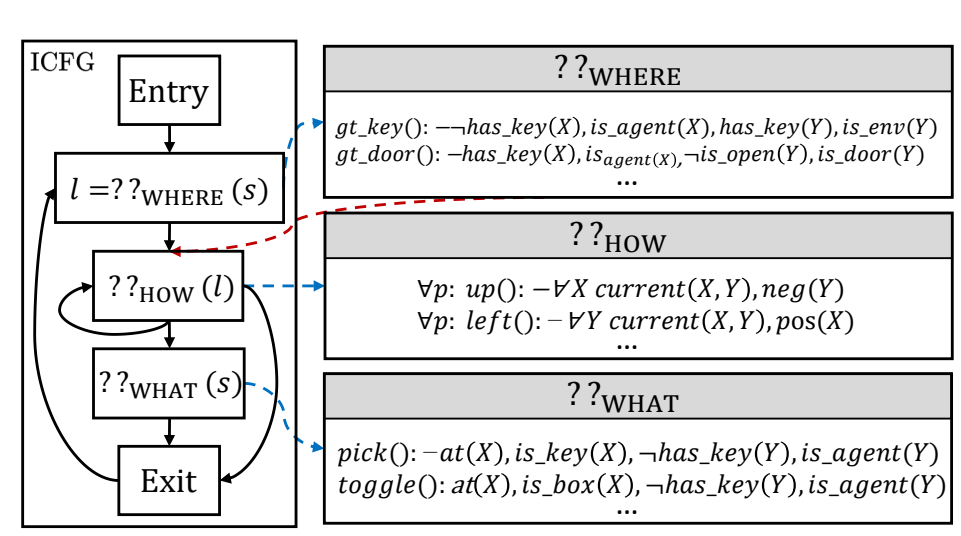

Figure 4: (left) A program sketch represented as ICFG and (right) the synthesized logic.

based on which sketching-based inductive logic programming is performed. Specifically, as shown in the inter-procedural control flow graph (ICFG) illustration Figure 4, the program sketch contains three major basic blocks of *hole functions* (denoted as $\rho$) to be synthesized: $??_{\text{WHERE}}$, $??_{\text{HOW}}$ and $??_{\text{WHAT}}$.

During each round of recursion, the program first executes and checks whether termination condition is satisfied, if not, an assignment statement is executed: $\boldsymbol{l} := ??_{\text{WHERE}}(\boldsymbol{s})$ by calling a *hole function*: $??_{\text{WHERE}}$. We define the meaning of *hole function* following the formalism of standard denotational semantics [32] in Figure 3(b). Concretely, for $??_{\text{WHERE}}$, the body of the synthesized clause $\mathcal{A}_{\text{WHERE}}(\boldsymbol{s})$ is constructed from the Herbrand base representation of the current state $\boldsymbol{s}$, namely objects' states and agent's attributes: $G_{\text{WHERE}} = \{\psi_j(obj_i) : i \in [1, m], j \in [1, n]\} \cup \{\psi_y(attr_x) : x \in [1, u], y \in [1, v]\}$. The clause body entails the head atom $g$, which denotes an abstract object within the environment (i.e., a sub-goal that agent shall arrive during this round of recursion, e.g., *key*, *box*, etc.). The semantic function $\mathcal{C}[\![\cdot]\!]$ evaluates the clause and returns the relative coordinates between the agent and the subgoal. The return value is passed to the logic sub-program $??_{\text{HOW}}$ (shown as the red dashed arrow). $??_{\text{HOW}}$ deduces the direction $d$ of next time step the agent shall move to: $pos \mapsto \mathcal{C}[\![\mathcal{A}_{\text{HOW}}(@) \models d]\!]\boldsymbol{s}$, where $pos$ is the agent's next-time-step position after execution, $\mathcal{A}_{\text{HOW}}(@)$ is constructed from Herbrand base which consists of ground atoms that applies predicates regarding numerical feature on the relative coordinates: $G_{\text{HOW}} = \{\psi_i(x), \psi_i(y) : i = [1, n]\}$. $??_{\text{HOW}}$ executes recursively until the sub-goal specified by $??_{\text{WHERE}}$ is achieved. Finally, $??_{\text{WHAT}}$ deduces the action $a$ to take to interact with the object at the sub-goal position: $o \mapsto \mathcal{C}[\![\mathcal{A}_{\text{WHAT}}(\boldsymbol{s}) \models a]\!]\boldsymbol{s}$, where $o$ denotes the updated state of the interacted object. Note that the program sketch we used is generalizable and can be applied to environments with different logic (see details in Section 4). For tasks whose environments are significantly different from the ones evaluated in this work, modifying or redesigning the sketch is also straightforward [38, 52].

### 3.3 Sketch-based Program Synthesis

GALOIS interacts with the environment to collect experiences to synthesize white-box programs with hierarchical and *cause-effect* logic following $\mathcal{L}_{\text{hybrid}}$. As shown in Figure 2(b), in the following, we illustrate how the program interacts with the environment, what is the structure of the program and how it is trained.

Practically, different from the black-box model, GALOIS requires different types of input and output. Therefore, GALOIS maintains an encoder $E(\cdot)$ and a decoder $D(\cdot)$ to interact with the environment. $E(s)$ maps the state $s$ to a set of ground atoms (formatted as valuation vector $\mathbf{e}_\rho$) with the verification from $\mathcal{L}_{\text{hybrid}}$, i.e., $\mathbf{e}_\rho = E(s, \mathcal{L}_{\text{hybrid}})$). As shown in Figure 2(b), the leftmost squares with different color represents the atoms from different hole functions (e.g., blue squares $\{f^{d=1,t}_{\text{WHERE}}\}^m_{t=1}$ denotes the atoms for $??_{\text{WHERE}}$ ($d$ denotes $d-1$ forward-chaining steps performed)). Based on $\mathbf{e}_\rho$, GALOIS outputs predicate probabilities and the Decoder maps them to the action probabilities (i.e., $a \sim D(p(\mathbf{e}_\rho))$, where $p(\cdot)$ denotes the deduction process).

In this way, the program is executable via fuzzy conjunction [11, 12], and the program synthesis can be performed. Guided by the semantics of the hole functions, GALOIS performs deduction using the weights $\boldsymbol{\theta}$ assigned to each candidate clauses of the specific atom (i.e., one weight $\theta$ in the weights vector $\boldsymbol{\theta}$ indicates one candidate clause). This process is shown in Figure 2(b). The rightmost squares represent the final atom deduced in the corresponding hole function. GALOIS combines all the ground atoms to perform a complex program. For example, $f^{2,1}_{\text{WHERE}}$ is inferred with conjunction between $f^{1,1}_{\text{WHERE}}$ and $f^{1,2}_{\text{WHERE}}$. A learnable weight is assigned to each candidate clause (e.g., $\theta^{3,1}_{\text{WHERE}}$ associates with the clause : `gt_key():- ¬has_key(X), is_agent(X), has_key(Y), is_env(Y)` which is derived with two steps of deduction, shown in Figure 4).

Now we explain in detail how a certain predicate is deduced. Given initialized valuation vector set $\mathbf{e}_\rho$, the deductions of the predicates are:

$$p(\mathbf{e}^\tau_\rho; \boldsymbol{\theta}) = \mathbf{e}^{\tau-1}_\rho \oplus (\sum_\psi \text{softmax}(\boldsymbol{\theta}^{\tau,\psi}_\rho) \odot h(\mathbf{e}^{\tau-1,\psi}_\rho)), \psi \in \Psi^{h(t)},$$

where $\boldsymbol{e}^\tau_\rho$ denotes the valuation vector for all the atoms in hole function $\rho$ at deduction step $\tau$ (initialized to 1), which is essentially a vector that stores the inferred truth values for all the corresponding atoms. $\oplus$ denotes the probabilistic sum: $\mathbf{x} \oplus \mathbf{y} = \mathbf{x} + \mathbf{y} - \mathbf{x} \cdot \mathbf{y}$. Specifically, given the normalized

(a) DoorKey
(b) BoxKey
(c) Unlockpickup
(b) Multiroom

Figure 5: Visualization of various tasks in MiniGrid, each requires different logic to accomplish: (a) the (red triangle) agent aims to pick up the (yellow) key to open the door (yellow box) and move to the goal (in green); (b) the agent needs to open the (gray) box to get the key first, then open the door to reach the goal; (c) the agent has to pick up the key to open the door, and then drop the key to pick up the (green) box; (d) the agent need to open multiple (yellow, blue, and red) doors to reach the goal.

weight vector $\boldsymbol{\theta}_\rho^{\tau,\psi}$ for the predicate $\psi$ in hole function $\rho$ at deduction step $\tau$, to perform a single-step deduction, we take the Hadamard product $\odot$ of $\boldsymbol{\theta}_\rho^{\tau,\psi}$ and the one-step inference results based on the valuation vector of last forward-chaining step, where $h(\cdot)$ denotes the inference function [4]. We then obtain the deductive result by taking the sum of all the intentional predicates. Finally, the valuation vector is updated to be $\boldsymbol{e}_\rho^\tau$ by taking the probabilistic sum $\oplus$ of the deductive result and the last step valuation vector. Intuitively, this process is similar to the forward propagation of a neural network, while GALOIS uses logic deduction to generate results.

The policy is trained in an on-policy manner. For each episode, the RL agent collects experiences $\{(s_t, a_t, r_t)\}_{t=0}^T$ by interacting with the environment using current policy $\pi_\theta$. With the collected experiences, GALOIS can thus synthesize the optimal hierarchical logic program to get the maximum expected cumulative return: $\pi_{\theta^*} = \arg\max_\theta \mathbb{E}_{a\sim\pi_\theta}\left[\sum_{t=0}^T \gamma^t r(s_t, a_t)\right]$, where $\theta$ denotes the learnable parameters in GALOIS. We train it with the Monte-Carlo policy gradient [42]:

$$\theta' = \theta + \alpha \nabla_\theta \log \pi_\theta Q_{\pi_\theta}(s_t, a_t) + \gamma_\epsilon \nabla_\theta H(\pi_\theta).$$

where $H(\pi_\theta)$ is the entropy regularization to improve exploration [22], the $\gamma_\epsilon$ is the hyperparameter to control the decrease rate of the entropy with time.

# 4 Experiments

To evaluate the effectiveness of GALOIS, we study the following research questions (RQs):
**RQ1 (Performance):** How effective is GALOIS regarding the performance and learning speed?
**RQ2 (Generalizability):** How is the generalizability of GALOIS across environments?
**RQ3 (Reusability):** Does GALOIS show great knowledge reusability across different environments?

## 4.1 Setup

**Environments:** We adopt the MiniGrid environments [7], which contains various tasks that require different abilities (i.e., navigation and multistep logical reasoning) to accomplish. We consider four representative tasks with incremental levels of logical difficulties as shown in Figure 5.
**Baselines:** Various baseline are used for comparisons, including mainstream DRL approaches, i.e., value-based (DQN [23]), policy-based (PPO [31]), actor-critic (SAC [13]), hierarchical (h-DQN [20]) algorithms, and the program synthesis guided methods (MPPS [44]). To avoid unfair comparison, we use the same training settings for all methods (see Appendix B for more details).

## 4.2 Performance Analysis (RQ1).

To answer RQ1, we evaluate the performance of GALOIS and other baseline methods in the training environment. The results in Figure 6 show that GALOIS outperforms all other mainstream baselines in terms of performance in environments that require complex logic, showing that GALOIS can learn

[4]Please refer to the $F_c$ function in the original paper [12] for specific details.

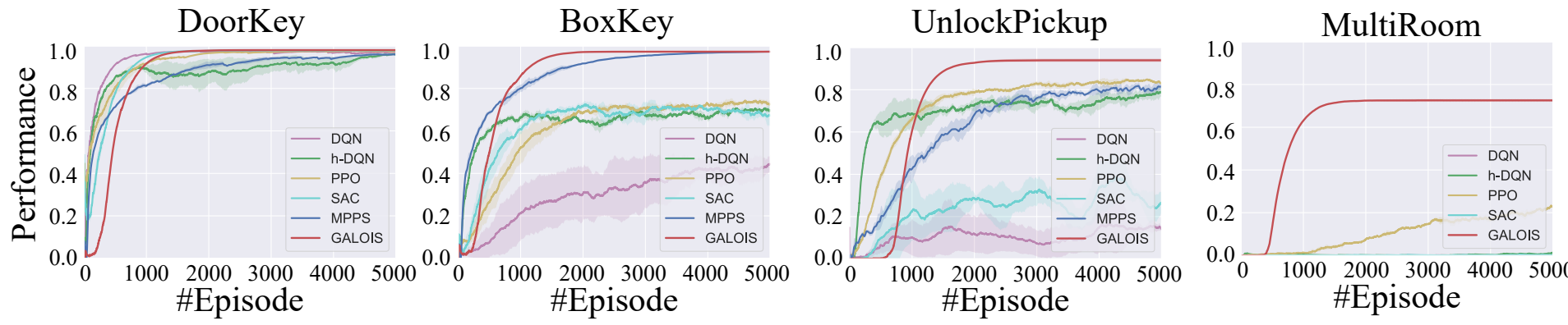

Figure 6: Comparisons of GALOIS and related baselines regarding the asymptotic performance and learning speed (all the results are averaged over 5 random seeds).

Table 1: Average return on the training environment and corresponding test environments with different sizes, *(v)* denotes agent trained with valuation vectors, (tr) denotes the training environment.

| | Size ($n$) | DQN | DQN(v) | SAC | SAC(v) | PPO | PPO(v) | hDQN | hDQN(v) | MPPS | MPPS(v) | Ours(v) |
|---|---|---|---|---|---|---|---|---|---|---|---|---|
| DoorKey | 8*8 (tr) | 0.473±0.130 | 0.919±0.071 | 0.966±0.019 | 0.938±0.052 | 0.919±0.017 | 0.958±0.008 | 0.979±0.002 | 0.928±0.114 | 0.861±0.046 | 0.949±0.021 | 0.963 ±0.008 |
| | 10*10 | 0.166±0.072 | 0.794±0.170 | 0.791±0.133 | 0.818±0.136 | 0.717±0.024 | 0.871±0.028 | 0.452±0.496 | 0.834±0.335 | 0.894±0.022 | 0.941±0.033 | 0.963 ±0.007 |
| | 12*12 | 0.050±0.035 | 0.730±0.175 | 0.527±0.066 | 0.829±0.184 | 0.494±0.021 | 0.800±0.014 | 0.152±0.263 | 0.769±0.232 | 0.906±0.029 | 0.950±0.040 | 0.963 ±0.007 |
| | 14*14 | 0.028±0.022 | 0.698±0.109 | 0.362±0.044 | 0.787±0.132 | 0.403±0.056 | 0.726±0.008 | 0.000±0.000 | 0.734±0.251 | 0.904±0.027 | 0.952±0.040 | 0.965 ±0.006 |
| | 16*16 | 0.006±0.005 | 0.877±0.109 | 0.161±0.081 | 0.886±0.065 | 0.269±0.035 | 0.750±0.008 | 0.000±0.000 | 0.755±0.235 | 0.910±0.047 | 0.944±0.045 | 0.963 ±0.007 |
| | 18*18 | 0.000±0.000 | 0.680±0.238 | 0.149±0.071 | 0.734±0.173 | 0.139±0.032 | 0.543±0.021 | 0.000±0.000 | 0.799±0.214 | 0.911±0.050 | 0.932±0.033 | 0.964 ±0.005 |
| | 20*20 | 0.000±0.000 | 0.746±0.184 | 0.099±0.042 | 0.690±0.185 | 0.211±0.062 | 0.768±0.034 | 0.000±0.000 | 0.729±0.256 | 0.929±0.028 | 0.963±0.007 | 0.966 ±0.005 |
| BoxKey | 8*8 (tr) | 0.241±0.166 | 0.305±0.112 | 0.608±0.046 | 0.711±0.041 | 0.643±0.029 | 0.714±0.051 | 0.488±0.273 | 0.541±0.056 | 0.864±0.069 | 0.949±0.003 | 0.975 ±0.001 |
| | 10*10 | 0.072±0.012 | 0.262±0.091 | 0.610±0.098 | 0.767±0.064 | 0.564±0.076 | 0.769±0.015 | 0.359±0.285 | 0.478±0.028 | 0.882±0.065 | 0.946±0.010 | 0.981 ±0.001 |
| | 12*12 | 0.007±0.012 | 0.256±0.035 | 0.411±0.084 | 0.830±0.014 | 0.470±0.117 | 0.844±0.044 | 0.302±0.227 | 0.604±0.042 | 0.881±0.088 | 0.950±0.000 | 0.985 ±0.000 |
| | 14*14 | 0.000±0.000 | 0.237±0.035 | 0.235±0.054 | 0.844±0.040 | 0.340±0.074 | 0.816±0.052 | 0.231±0.132 | 0.507±0.084 | 0.893±0.090 | 0.952±0.008 | 0.987 ±0.000 |
| | 16*16 | 0.007±0.012 | 0.290±0.045 | 0.206±0.062 | 0.846±0.052 | 0.254±0.054 | 0.835±0.027 | 0.198±0.124 | 0.607±0.014 | 0.861±0.155 | 0.958±0.001 | 0.988 ±0.000 |
| | 18*18 | 0.000±0.000 | 0.224±0.023 | 0.131±0.042 | 0.863±0.005 | 0.155±0.084 | 0.846±0.075 | 0.099±0.105 | 0.568±0.014 | 0.879±0.120 | 0.963±0.002 | 0.990 ±0.000 |
| | 20*20 | 0.000±0.000 | 0.251±0.046 | 0.071±0.035 | 0.844±0.022 | 0.124±0.038 | 0.874±0.042 | 0.093±0.011 | 0.463±0.127 | 0.905±0.097 | 0.957±0.013 | 0.987 ±0.009 |
| UnlockPickup | 6*6 (tr) | 0.236±0.240 | 0.428±0.164 | 0.222±0.069 | 0.510±0.145 | 0.763±0.014 | 0.826±0.054 | 0.496±0.346 | 0.824±0.233 | 0.645±0.104 | 0.813±0.039 | 0.901 ±0.021 |
| | 8*8 | 0.008±0.017 | 0.324±0.159 | 0.164±0.059 | 0.457±0.196 | 0.578±0.094 | 0.869±0.023 | 0.187±0.225 | 0.820±0.257 | 0.747±0.143 | 0.872±0.025 | 0.933 ±0.014 |
| | 10*10 | 0.000±0.000 | 0.307±0.122 | 0.080±0.020 | 0.460±0.252 | 0.364±0.112 | 0.908±0.010 | 0.097±0.164 | 0.843±0.208 | 0.765±0.115 | 0.935±0.012 | 0.953 ±0.007 |
| | 12*12 | 0.000±.0.000 | 0.263±0.216 | 0.042±0.012 | 0.488±0.253 | 0.198±0.045 | 0.902±0.009 | 0.051±0.102 | 0.822±0.271 | 0.802±0.080 | 0.936±0.002 | 0.957 ±0.011 |
| | 14*14 | 0.000±0.000 | 0.277±0.233 | 0.021±0.019 | 0.472±0.318 | 0.176±0.039 | 0.919±0.014 | 0.024±0.053 | 0.869±0.192 | 0.834±0.075 | 0.962±0.003 | 0.969 ±0.004 |
| | 16*16 | 0.000±0.000 | 0.231±0.171 | 0.018±0.022 | 0.496±0.305 | 0.128±0.053 | 0.876±0.030 | 0.012±0.026 | 0.800±0.318 | 0.841±0.122 | 0.961±0.010 | 0.973 ±0.005 |
| | 18*18 | 0.000±0.000 | 0.205±0.146 | 0.003±0.005 | 0.470±0.317 | 0.032±0.032 | 0.899±0.049 | 0.000±0.000 | 0.827±0.273 | 0.870±0.062 | 0.963±0.009 | 0.977 ±0.000 |
| Multiroom | 8*8 (tr) | 0.000±0.000 | 0.014±0.008 | 0.000±0.000 | 0.007±0.007 | 0.002±0.003 | 0.236±0.036 | 0.000±0.000 | 0.000±0.000 | N/A | N/A | 0.663 ±0.018 |
| | 10*10 | 0.000±0.000 | 0.000±0.000 | 0.000±0.000 | 0.000±0.000 | 0.000±0.000 | 0.166±0.026 | 0.000±0.000 | 0.000±0.000 | N/A | N/A | 0.622 ±0.017 |
| | 12*12 | 0.000±0.000 | 0.000±0.000 | 0.000±0.000 | 0.000±0.000 | 0.000±0.000 | 0.115±0.050 | 0.000±0.000 | 0.000±0.000 | N/A | N/A | 0.607 ±0.012 |
| | 14*14 | 0.000±0.000 | 0.000±0.000 | 0.000±0.000 | 0.000±0.000 | 0.000±0.000 | 0.072±0.030 | 0.000±0.000 | 0.000±0.000 | N/A | N/A | 0.529 ±0.020 |
| | 16*16 | 0.000±0.000 | 0.000±0.000 | 0.000±0.000 | 0.000±0.000 | 0.000±0.001 | 0.100±0.009 | 0.000±0.000 | 0.000±0.000 | N/A | N/A | 0.596 ±0.015 |
| | 18*18 | 0.000±0.000 | 0.000±0.000 | 0.000±0.000 | 0.000±0.000 | 0.000±0.000 | 0.074±0.028 | 0.000±0.000 | 0.000±0.000 | N/A | N/A | 0.529 ±0.029 |
| | 20*20 | 0.000±0.000 | 0.000±0.000 | 0.000±0.000 | 0.000±0.000 | 0.001±0.003 | 0.078±0.037 | 0.000±0.000 | 0.000±0.000 | N/A | N/A | 0.519 ±0.023 |

the comprehensive task-solving logic, leading to the highest performance. Note that in the DoorKey environment, all baseline methods can reach optimal training performance, and DQN converges the fastest. This is because the DoorKey environment is relatively simpler, whose intrinsic logic is easy to learn, and hierarchical models have more parameters than the DQN model, leading to a slower convergence speed. Moreover, we observe that MPPS, hDQN, and GALOIS converge faster than the methods without hierarchy in environments that require more complex logic (e.g., UnlockPickup, BoxKey). We can thus conclude that introducing hierarchy contributes to more efficient learning. Besides, unlike other pure neural network baselines, GALOIS and MPPS present steadier asymptotic performance during training with also smaller variance. This result demonstrates the effectiveness of introducing program synthesis for steady policy learning.

Specifically, MPPS theoretically fails on MultiRoom as there exists no deterministic imperative program description (denoted as N/A in Table 2). The reason is that the sequence of colored doors that the agent should cross differs for each episode (e.g., ep1: `red_door`→`yellow_door`→`blue_door`, ep2: `blue_door`→`red_door`→`yellow_door`), thus the program on solving this task is dynamically changing, which fails the imperative program synthesizer. This further indicates the importance of synthesizing declarative programs with *cause-effect* logic instead of merely finding the ordered sequence of subprograms for solving tasks. More details are discussed in the following sections.

### 4.3 Generalizability Analysis (RQ2).

To answer RQ2, we evaluate models' performance on test environments with different sizes and task-solving logic (i.e., semantic modifications). Concretely, as shown in Table 1, GALOIS outperforms all the other baseline methods (highest average returns are highlighted in gray) and maintains near-optimal performance. Furthermore, we also observe that all other baseline methods maintain acceptable generalizability. This contradicts the conclusion in [16] that neural network-based agents fail to generalize to environments with size changes. We hypothesize that this attributes to the use

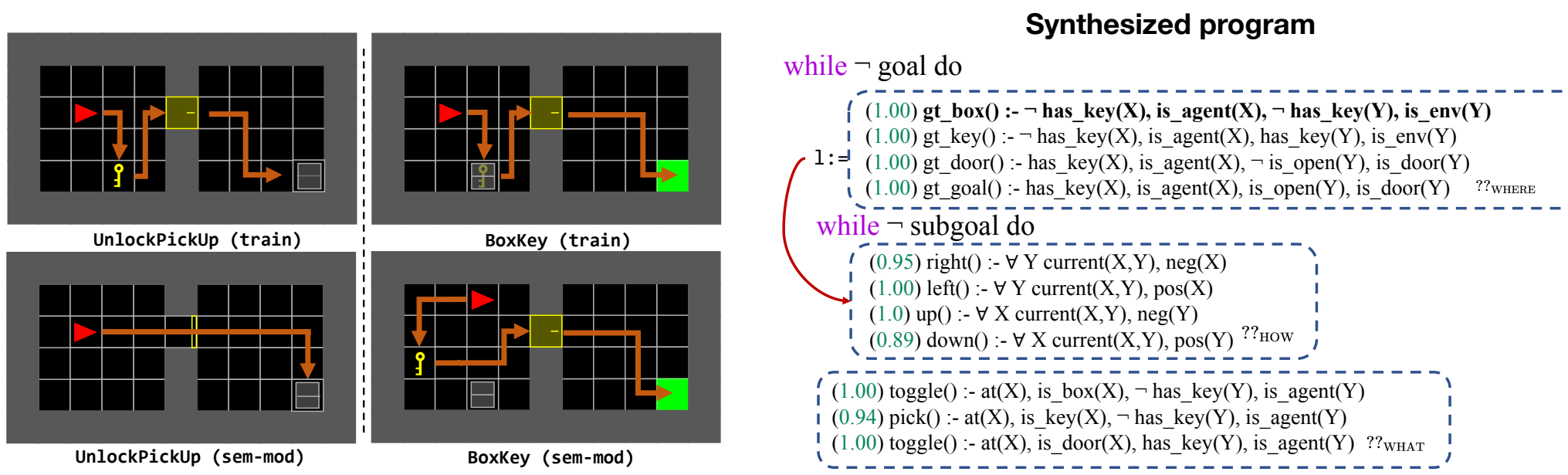

Figure 7: (left) shows the original and semantic-modified environments of UnlockPickup and BoxKey. The optimal traces are marked in orange; (right) shows the synthesized program for BoxKey.

Table 2: Average return on test environments with semantic modifications.

| | | DQN | SAC | PPO | hDQN | MPPS | Ours |
|---|---|---|---|---|---|---|---|
| BoxKey | 8*8(tr) | 0.241±0.166 | 0.608±0.046 | 0.714±0.042 | 0.541±0.056 | 0.949±0.003 | 0.975 ±0.001 |
| | *sem-mod* | 0.040±0.040 | 0.098±0.005 | 0.126±0.008 | 0.476±0.091 | 0.119±0.020 | 0.976 ±0.001 |
| UnlockPickup | 12*6(tr) | 0.236±0.240 | 0.222±0.069 | 0.826±0.054 | 0.824±0.233 | 0.813±0.039 | 0.901 ±0.021 |
| | *sem-mod* | 0.007±0.012 | 0.040±0.005 | 0.098±0.004 | 0.434±0.390 | 0.000±0.000 | 0.983 ±0.003 |

of different types of representations. To evaluate the effectiveness of using the valuation vector representation, we conduct experiments using the observations directly obtained from environments (e.g., the status and locations of objects). Surprisingly, though achieving decent performance in the training environment, all the vanilla neural network-based baselines perform poorly on test environments of different sizes. Therefore, we conclude that by introducing logic expression as state representation (in the form of valuation vectors), better generalizability can be obtained. However, as illustrated by the results, the valuation vector itself is not enough to achieve supreme generalizability, GALOIS manages to achieve even better generalizability due to explicit use of *cause-effect* logic with a hierarchical structure.

We then evaluate models' generalizability on two test environments with minor semantic modifications, namely BoxKey (*sem-mod*) and UnlockPickup (*sem-mod*), as shown in Figure 7 (left). Specifically, for UnlockPickup (*sem-mod*), different from the training environment, there is no key in the environment, and the door is already open. And thus the agent should head for the target location directly. For BoxKey (*sem-mod*), the key is placed outside the box. Thus, optimally, the agent should directly head for the key and ignore the existence of the box. The results in Table 2 indicate that all the baselines are severely compromised while GALOIS retains near-optimal generalizability. This attributes to its explicit use of *cause-effect* logic.

Figure 7 (right) shows an example synthesized program of GALOIS (we include more synthesized program examples in Appendix A). *E.g.* The program specifies the cause of `gt_box()` (marked in bold) as the agent has no key and there exists no visible key in the environment. Thus when placed under BoxKey(*sem-mod*), GALOIS agent would skip `gt_box()` and head directly for the key since `gt_box()` is grounded as false by the logic clause body. The result indicates that the explicit use of *effect-effect* logic is not only verifiable for humans but allows GALOIS model to perform robustly in environments with different task-solving logic. For MPPS, since it only learns a fixed sequence of sub-programs it fails to generalize. *E.g.* the synthesized program of MPSS trained on BoxKey is: `toggle_box();get_key();open_door();reach_goal()`, thus when the key is placed under BoxKey(*sem-mod*), the agent would follow the learned program and redundantly toggle the box first.

### 4.4 Knowledge Reusability Analysis (RQ3)

To answer RQ3, we initialize a GALOIS model's weights with the knowledge base learned from other tasks (e.g., DoorKey→BoxKey) and fine-tune the entire model continuously. Figure 8 shows the detailed results of knowledge reusability among three different environments. Apparently, the learning efficiency can be significantly increased by warm-starting the weights of the GALOIS model with knowledge learned from different tasks with overlapped logic compared with the one that is

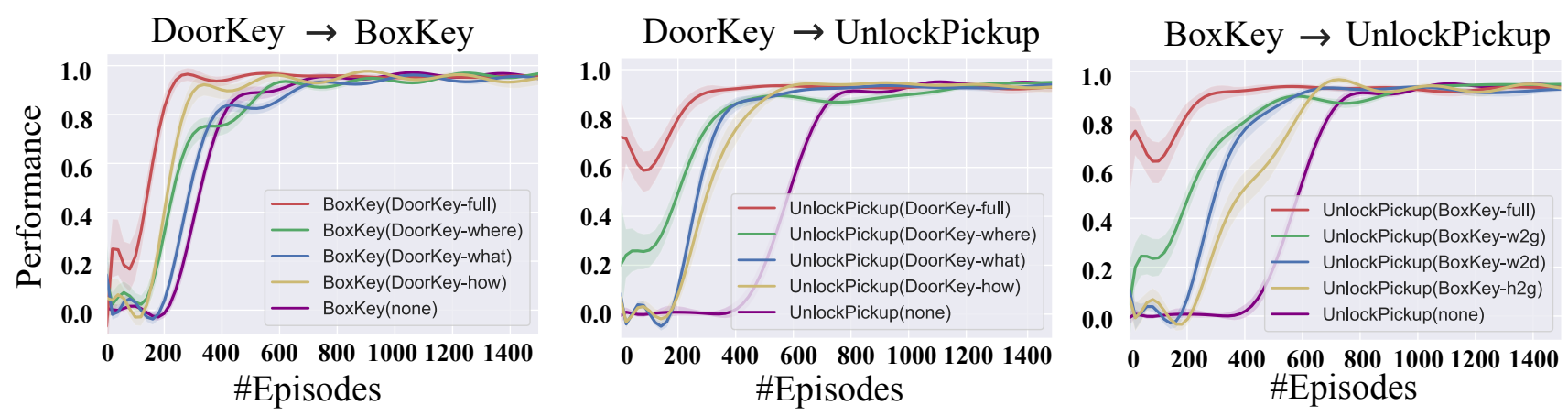

Figure 8: Knowledge reusability across different environments. `full` denotes warm-starting policy with the full program, {`where`, `how`, `what`} denotes warm-starting with only the sub-program from the corresponding hole function (e.g., $??_{\text{WHERE}}$), `none` means learning from scratch.

learned from scratch. Furthermore, we demonstrate the reusability of knowledge from each sub-program, respectively. The results show that a considerable boost in learning efficiency can already be obtained by reusing knowledge from each sub-program respectively (e.g., *BoxKey(DoorKey-where)* agent is only warm-started with the sub-program of $??_{\text{WHERE}}$), which is an advantage brought by GALOIS's hierarchical and *cause-effect* logic. Figure 9 shows an example of knowledge reusing from DoorKey to BoxKey environments (*BoxKey(DoorKey-full)*). By reusing the logic learned from the DoorKey environment (the orange path in Figure 9), agent only needs to learn the *cause-effect* logic of `toggle_box()` from scratch, which greatly boosts the learning efficiency.

## 5 Related Work

**Neural Program Synthesis:** Given a set of program specifications (e.g., I/O examples, natural language instructions, etc.), program synthesis aims to induce an explicit program that satisfies the given specification. Recent works illustrate that neural networks are effective in boosting both the synthesis accuracy and efficiency [9, 6, 3, 12, 18]. Devlin et al. [9] propose using a recurrent neural network to synthesize programs for string transformation. Chen et al. [6] further proposes incorporating intermediate execution results to augment the model's input state, which significantly improves performance for imperative program synthesis. ∂ILP [12] proposes modeling the forward chaining mechanism with a statistical model to achieve synthesis for Datalog programs.

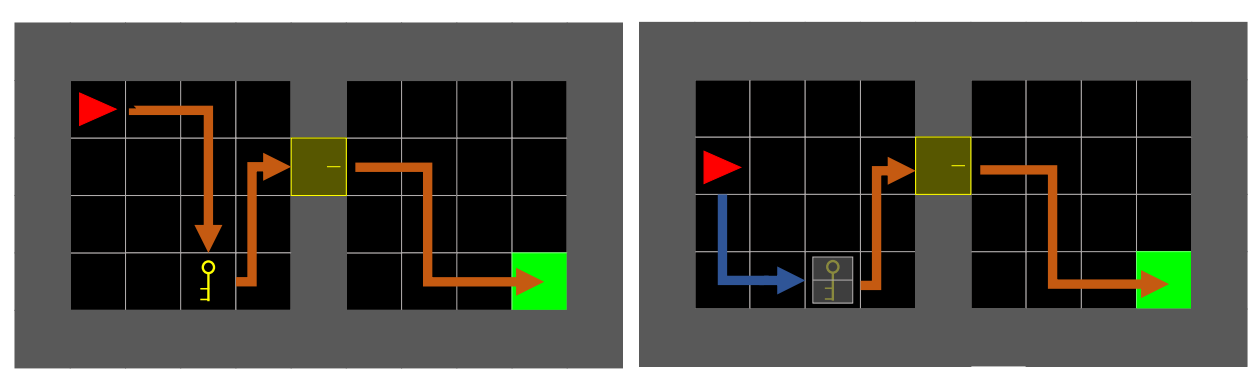

(a) DoorKey (b) BoxKey

Figure 9: The illustration of knowledge reused from DoorKey to BoxKey. The orange path represents the reusable knowledge (learned from DoorKey and directly reused in BoxKey).

**Program Synthesis by Sketching:** Many real-world synthesis problems are intractable, posing a great challenge for the synthesis model. *Sketching* [38, 52, 37] is a novel program synthesis paradigm that proposes establishing the synergy between the human expert and the synthesizer by embedding domain expert knowledge as general program sketches (i.e., a program with unspecified fragments to be synthesized), based on which the synthesis is conducted. Singh et al. [37] propose a feedback generation system that automatically synthesizes program correction based on a general program sketch. Nye et al. [25] propose a two-stage neural program synthesis framework that first generates a coarse program sketch using a neural model, then leverages symbolic search for second-stage fine-tuning based on the generated sketch.

**Program Synthesis for Reinforcement Learning:** Leveraging program synthesis for the good of reinforcement learning has been increasingly popular as it is demonstrated to improve performance and interpretability significantly. Jiang et al. [16] introduce using ∂ILP model for agent's policy, which improves downstream generalization by expressing policy as explicit functional programs. Imperative programs are used as a novel implementation of hierarchical reinforcement learning in which the agent's policy is guided by high-level programs [40, 44, 15]. In addition, program synthesis has also been used as a post hoc interpretation method for neural policies [41, 2].

## 6 Conclusion

In this work, we propose a novel generalizable logic synthesis framework GALOIS that can synthesize programs with hierarchical and *cause-effect* logic. A novel sketch-based DSL is introduced to allow hierarchical logic programs. Based on that, a hybrid synthesis method is proposed to synthesize programs with generalizable *cause-effect* logic. Experimental results demonstrate that GALOIS can significantly outperform DRL and previous program-synthesis-based methods in terms of learning ability, generalizability, and interpretability. Regarding limitation, as it is general for all program synthesis-based methods, the input images need to be pre-processed into Herbrand base for the synthesis model, which is required to be done once for each domain. Therefore, automatic Herbrand base learning would be an important future direction. Another promising direction is applying GALOIS in more challenge competitive multi-agent scenarios [50, 14, 47] or cooperative multi-agent scenarios [21, 39, 49]. We state that our work would not produce any potential negative societal impacts.

## Acknowledgments and Disclosure of Funding

This work is supported by the National Natural Science Foundation of China (Grant No. 62106172, U1836214), New Generation of Artificial Intelligence Science and Technology Major Project of Tianjin (Grant No.: 19ZXZNGX00010), Science and Technology on Information Systems Engineering Laboratory (Grant No. WDZC20205250407), National Research Foundation, Prime Ministers Office, Singapore under its National Cybersecurity R&D Program (Award No. NRF2018NCR-NCR005-0001), NRF Investigatorship NRF-NRFI06-2020-0001, the Ministry of Education, Singapore under its Academic Research Fund Tier 1 (21-SIS-SMU-033), Tier 2 (MOE2019-T2-1-040), and Tier 3 (MOET32020-0004). Any opinions, findings and conclusions or recommendations expressed in this material are those of the author(s) and do not reflect the views of the Ministry of Education, Singapore.

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
