# GALOIS: Boosting Deep Reinforcement Learning via Generalizable Logic Synthesis

**Yushi Cao**[2,*], **Zhiming Li**[2,*], **Tianpei Yang**[1,3,†], **Hao Zhang**[1], **Yan Zheng**[1,†]
**Yi Li**[2], **Jianye Hao**[1], **Yang Liu**[2]

[1]College of Intelligence and Computing, Tianjin university, Tianjin, China
[2]Nanyang Technological University, Singapore, [3]University of Alberta, Canada

{yushi002,zhiming001}@e.ntu.edu.sg
{tpyang,3018216216,yanzheng,jianye.hao}@tju.edu.cn
{yi_li,yangliu}@ntu.edu.sg

## A   Synthesized Programs

Figure 1: Detailed synthesized programs.

In this section, we present the detailed synthesized programs for the four evaluated environments in Figure 1.

**DoorKey**   As shown in Figure 1a, regarding the hole function $??_{\text{WHERE}}$, the agent goes to the key if it verifies that the grounded fact to be true: the agent has no key and the door is closed; goes to door if it has key and the door is closed; goes to target location if it has key and the door is already open. And for hole function, $??_{\text{WHAT}}$, the agent picks up the key if it is adjacent to the key and it has no key; toggles the door if it is adjacent to the door and it has the key.

---

*Equal contribution.
†Corresponding authors: Yan Zheng (yanzheng@tju.edu.cn) and Tianpei Yang (tpyang@tju.edu.cn).

36th Conference on Neural Information Processing Systems (NeurIPS 2022).

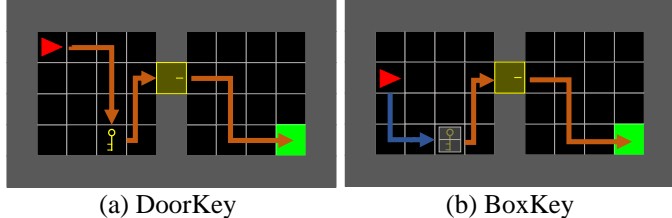

|   (a) DoorKey   |   (b) BoxKey   |

Figure 2: The illustration of knowledge reused from DoorKey to BoxKey. The orange path in (b) represents the reusable knowledge from (a). The blue path in (b) denotes the new knowledge need to learn for BoxKey.

**BoxKey**   As shown in Figure 1b, different from DoorKey, it has to open the box to get the key. Thus the newly learned sub-program for hole function $??_{\text{WHERE}}$ would be: agent should go to the box if there is no key in the environment and the agent does not have a key. Accordingly, for $??_{\text{WHAT}}$, the agent would toggle (open) the box when it arrives at the box.

**UnlockPickup**   As shown in Figure 1c, since it has to pick up an object located at the target location to accomplish the task, the newly learned sub-program $??_{\text{WHERE}}$ would be: agent would drop the key when it has the key and the door is already open. And for $??_{\text{WHAT}}$, whenever the agent arrives at the goal position, it would pick up the box if it does not has a key and the door is already open.

**MultiRoom**   As shown in Figure 1d, regarding $??_{\text{WHERE}}$, agent would head for a door of a specific color if it is reachable and closed, e.g., `gt_red() :- reachable(X),is_red(X), ¬ is_open(X)`. Thus the learned program is color-agnostic (i.e., the agent's policy would remain robust no matter how the colored doors sequence changes).

## B   Implementation Details

In this section we present the implementation details of all the baseline methods and our approach[3], along with the information of the hardware on which these models are trained.

**Environment settings.**   The valuation vector representations are fed to all the methods as input. Specifically, for the MPPS model, we directly provide the oracle program to save time for synthesizing programs, which could be considered as the upper bound performance of MPPS [3]. For the h-DQN method, we keep the high-level goal the same as MPPS and GALOIS, (i.e., {key, door, goal} for DoorKey; {box, key, door, goal} for BoxKey; {box, key, door, drop} for UnlockPickup; {red door, yellow door, blue door, goal} for MultiRoom).

The reward from the MiniGrid environment is sparse (i.e., only a positive reward will be given after completing the task), thus in order to motivate the agent to learn to solve the task using the least amount of actions, we follow previous works [1, 3] and apply the reward shaping mechanism. For every action taken, a negative reward of $-0.01$ will be given, a small positive reward of $+0.20$ will be given for achieving each sub-task (e.g., picking up the key), and a reward of $+1.00$ will be given for completing the task.

**Training settings.**   For all DNN-based baseline methods, we use a 2-layer multilayer perceptron (MLP) with 128 neurons in each layer. We train all the baseline methods and GALOIS with the Adam optimizer [2] with a 0.001 learning rate. We use a batch size of 256. The entropy coefficient $\gamma_\epsilon$ starts at 5 and decays at a factor of 10 for every 50 episodes. We conducted all experiments on a Ubuntu 16.04 server with 24 cores of Intel(R) Xeon(R) Silver 4214 2.2GHz CPU, 251GB RAM, and an NVIDIA GeForce RTX 3090 GPU.

---

[3]The code is available at: https://github.com/caoysh/GALOIS

## Figure 3

**(a) DoorKey**

while ¬ goal do

1:−
- (1.00) gt_key() :- ¬ has_key(X), is_agent(X), has_key(Y), is_env(Y)
- (1.00) gt_door() :- has_key(X), is_agent(X), ¬ is_open(Y), is_door(Y)
- (1.00) gt_goal() :- has_key(X), is_agent(X), is_open(Y), is_door(Y) ??WHERE

while ¬ subgoal do
- (0.95) right() :- ∀ Y current(X,Y), neg(X)
- (1.00) left() :- ∀ Y current(X,Y), pos(X)
- (1.0) up() :- ∀ X current(X,Y), neg(Y)
- (0.89) down() :- ∀ X current(X,Y), pos(Y) ??HOW

- (0.94) pick() :- at(X), is_key(X), ¬ has_key(Y), is_agent(Y)
- (1.00) toggle() :- at(X), is_door(X), has_key(Y), is_agent(Y) ??WHAT

**(b) BoxKey**

while ¬ goal do

1:−
- (1.00) **gt_box() :- ¬ has_key(X), is_agent(X), ¬ has_key(Y), is_env(Y)**
- (1.00) gt_key() :- ¬ has_key(X), is_agent(X), has_key(Y), is_env(Y)
- (1.00) gt_door() :- has_key(X), is_agent(X), ¬ is_open(Y), is_door(Y)
- (1.00) gt_goal() :- has_key(X), is_agent(X), is_open(Y), is_door(Y) ??WHERE

while ¬ subgoal do
- (0.95) right() :- ∀ Y current(X,Y), neg(X)
- (1.00) left() :- ∀ Y current(X,Y), pos(X)
- (1.0) up() :- ∀ X current(X,Y), neg(Y) ??HOW
- (0.89) down() :- ∀ X current(X,Y), pos(Y)

- (1.00) **toggle() :- at(X), is_box(X), ¬ has_key(Y), is_agent(Y)**
- (0.94) pick() :- at(X), is_key(X), ¬ has_key(Y), is_agent(Y)
- (1.00) toggle() :- at(X), is_door(X), has_key(Y), is_agent(Y) ??WHAT

Figure 3: Reused program from DoorKey to BoxKey. The subprograms marked in blue (in (b)) need to be learned for BoxKey, while other subprograms are directly reused from (a).

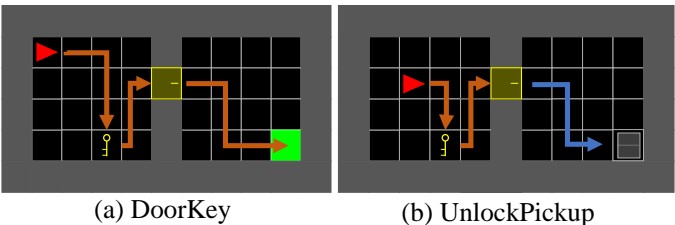

(a) DoorKey      (b) UnlockPickup

Figure 4: The illustration of knowledge reused from DoorKey to UnlockPickup. The orange path in (a) means knowledge learned from scratch. The orange path in (b) represents the reusable knowledge. The blue path in (b) means the new knowledge learned for UnlockPickup.

# C    Knowledge Reusability

In this section, we illustrate how the programs are reused from one environment to another in detail.

## C.1    DoorKey to BoxKey

Figure 2 illustrates the DoorKey and BoxKey environments and their optimal policies, respectively. Different from DoorKey, to complete the BoxKey, the agent has to open the box to get the key first, while the agent in DoorKey is able to get the key directly. Afterward, the optimal policies would be the same, i.e., pick up the key, open the door and then go to the goal. Concretely, the BoxKey agent will only need to learn how to go to the box and open it, and thus all the knowledge can be reusable from DoorKey. As shown in Figure 3b, the subprograms marked in blue are the new knowledge to learn to solve the BoxKey while other sub-programs can be directly reused from DoorKey.

## C.2    DoorKey to UnlockPickup

Figure 4 shows the DoorKey and UnlockPickup environments and their respective optimal policies. After opening the door, to solve UnlockPickup, the agent has to drop the key and then pick up the box. Specifically, the knowledge before dropping the key is reusable and the UnlockPickup agent continues to learn the rest. Due to the white-box nature of GALOIS, instead of reusing all the knowledge, the sub-program learned in DoorKey ( i.e., `gt_goal() :- has_key(X), is_agent(X), is_open(Y), is_door(Y)`) can be removed when reusing. As shown in Figure 5, the sub-programs marked in blue are the new knowledge to learn while others are directly reused from DoorKey.

## C.3    BoorKey to UnlockPickup

Figure 6 shows the BoorKey and UnlockPickup environments and their respective optimal policies. In specific, the agent in BoxKey learns the clause: `toggle():- at(X), is_box(X), ¬has_key(Y), is_agent(Y)` while in UnlockPickup, the clause should be: `pick():- at(X), is_box(X), ¬has_key(Y), is_agent(Y)`, as shown in Figure 7. This leads to a negative impact of reusing

## Figure 5

**(a) DoorKey**

while ¬ goal do

1:-
- (1.00) gt_key() :- ¬ has_key(X), is_agent(X), has_key(Y), is_env(Y)
- (1.00) gt_door() :- has_key(X), is_agent(X), ¬ is_open(Y), is_door(Y)
- (1.00) gt_goal() :- has_key(X), is_agent(X), is_open(Y), is_door(Y) ??WHERE

while ¬ subgoal do
- (0.95) right() :- ∀ Y current(X,Y), neg(X)
- (1.00) left() :- ∀ Y current(X,Y), pos(X)
- (1.0) up() :- ∀ X current(X,Y), neg(Y)
- (0.89) down() :- ∀ X current(X,Y), pos(Y) ??HOW

- (0.94) pick() :- at(X), is_key(X), ¬ has_key(Y), is_agent(Y)
- (1.00) toggle() :- at(X), is_door(X), has_key(Y), is_agent(Y) ??WHAT

**(b) UnlockPickup**

while ¬ goal do

1:-
- (1.00) gt_key() :- ¬ has_key(X), is_agent(X), has_key(Y), is_env(Y)
- (1.00) gt_door() :- has_key(X), is_agent(X), ¬ is_open(Y), is_door(Y)
- (1.00) gt_drop() :- has_key(X), is_agent(X), is_open(Y), is_door(Y)
- (1.00) gt_box() :- ¬ has_key(X), is_agent(X), is_open(Y), is_door(Y) ??WHERE

while ¬ subgoal do
- (0.95) right() :- ∀ Y current(X,Y), neg(X)
- (1.00) left() :- ∀ Y current(X,Y), pos(X)
- (1.0) up() :- ∀ X current(X,Y), neg(Y)
- (0.89) down() :- ∀ X current(X,Y), pos(Y) ??HOW

- (0.95) pick() :- at(X), is_key(X), ¬ has_key(Y), is_agent(Y)
- (0.94) drop() :- at(X), is_drop(X), has_key(Y), is_agent(Y)
- (1.00) toggle() :- at(X), is_door(X), has_key(Y), is_agent(Y)
- (0.99) pick() :- at(X), is_box(X), ¬ has_key(Y), is_agent(Y) ??WHAT

Figure 5: Reused program from DoorKey to UnlockPickup. The sub-programs marked in blue (in (b)) are needed to be learned for UnlockPickup while other sub-programs are directly reused from (a).

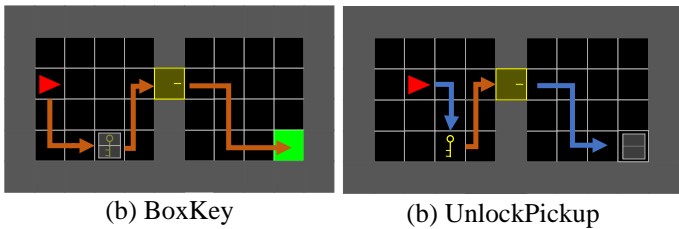

**(b) BoxKey**      **(b) UnlockPickup**

Figure 6: The illustration of knowledge reused from BoorKey to UnlockPickup. The orange path in (a) means knowledge learned from scratch. The orange path in (b) represents the reusable knowledge. The blue path in (b) means the new knowledge learned for UnlockPickup.

## Figure 7

**(a) BoorKey**

while ¬ goal do

1:-
- (1.00) gt_box() :- ¬ has_key(X), is_agent(X), ¬ has_key(Y), is_env(Y)
- (1.00) gt_key() :- ¬ has_key(X), is_agent(X), has_key(Y), is_env(Y)
- (1.00) gt_door() :- has_key(X), is_agent(X), ¬ is_open(Y), is_door(Y)
- (1.00) gt_goal() :- has_key(X), is_agent(X), is_open(Y), is_door(Y) ??WHERE

while ¬ subgoal do
- (0.95) right() :- ∀ Y current(X,Y), neg(X)
- (1.00) left() :- ∀ Y current(X,Y), pos(X)
- (1.0) up() :- ∀ X current(X,Y), neg(Y) ??HOW
- (0.89) down() :- ∀ X current(X,Y), pos(Y)

- (1.00) toggle() :- at(X), is_box(X), ¬ has_key(Y), is_agent(Y)
- (0.94) pick() :- at(X), is_key(X), ¬ has_key(Y), is_agent(Y)
- (1.00) toggle() :- at(X), is_door(X), has_key(Y), is_agent(Y) ??WHAT

**(b) UnlockPickup**

while ¬ goal do

1:-
- (1.00) gt_key() :- ¬ has_key(X), is_agent(X), has_key(Y), is_env(Y)
- (1.00) gt_door() :- has_key(X), is_agent(X), ¬ is_open(Y), is_door(Y)
- (1.00) gt_drop() :- has_key(X), is_agent(X), is_open(Y), is_door(Y)
- (1.00) gt_box() :- ¬ has_key(X), is_agent(X), is_open(Y), is_door(Y) ??WHERE

while ¬ subgoal do
- (0.95) right() :- ∀ Y current(X,Y), neg(X)
- (1.00) left() :- ∀ Y current(X,Y), pos(X)
- (1.0) up() :- ∀ X current(X,Y), neg(Y)
- (0.89) down() :- ∀ X current(X,Y), pos(Y) ??HOW

- (0.95) pick() :- at(X), is_key(X), ¬ has_key(Y), is_agent(Y)
- (0.94) drop() :- at(X), is_drop(X), has_key(Y), is_agent(Y)
- (1.00) toggle() :- at(X), is_door(X), has_key(Y), is_agent(Y)
- (0.99) pick() :- at(X), is_box(X), ¬ has_key(Y), is_agent(Y) ??WHAT

Figure 7: Reused program from BoorKey to UnlockPickup. The sub-programs marked in blue (in (b)) are needed to be learned for UnlockPickup while other sub-programs are directly reused from (a).

knowledge as in UnlockPickup environment, the box will disappear if toggled and the task will never be completed. The experiments demonstrate the ability of GALOIS to handle nonreusable knowledge. Additionally, thanks to its white-box nature, this rule can also be identified and removed manually to avoid negative effects.