# OpenReview forum: "GALOIS: Boosting Deep Reinforcement Learning via Generalizable Logic Synthesis"
_NeurIPS.cc/2022/Conference — NeurIPS 2022 Accept_

### Official Review · Reviewer_sTbQ · 2022-06-28

**Rating:** 6
**Confidence:** 5
**Soundness:** 3 good
**Presentation:** 3 good
**Contribution:** 3 good

**Summary:**

This paper proposes GALOIS, where the model learns a hierarchical logic program to represent the policy for solving a task. They first design a fixed program sketch represented as an inter-procedural control flow graph (ICFG), which includes 3 code blocks. The model needs to generate several definite clauses in each code block. The model is trained with policy gradient to maximize the expected cumulative return. They evaluate GALOIS on several tasks using the MiniGrid environment. Besides comparing to mainstream DRL methods, they also compare GALOIS to another program synthesis method that generates programs without conditional statements. Experimental results demonstrate that GALOIS generalizes better to test environments that share some similarities to the training environment.

**Questions:**

1. How large is the program search space for each task? From Figure 1 in the appendix, I feel that the atom set is specialized for each task, though sometimes the atom set is shared among similar tasks. Then for each definite clause in code blocks, actually the head atom and the atom list in the body can also be fixed, and the model simply needs to predict whether there is a negation for each atom. In this case, the comparison to MPPS is a bit unfair, given that MPPS does not leverage a program sketch that already contains conditionals and loops.

2. In Section 4.2, I think the generalization capability largely comes from the design of subgoals, as shown in Figure 7. Related to my first question, I wonder how these subgoals are predicted in the first code block, because some of them are unnecessary if only considering the training environment.

3. How would GALOIS work on other benchmarks, e.g., those evaluated in the MPPS paper?

**Limitations:**

The authors adequately addressed the limitations and potential negative societal impact of their work.

**Strengths And Weaknesses:**

Program synthesis for learning RL policies is an interesting and important problem. In particular, representing the policy as a hierarchical logic program has a lot of benefits, including generalizability, interpretability, and potentially solving more complicated tasks. The program sketch and DSL designed in the GALOIS framework are well-suited for experiments in this work, and the authors conduct an extensive comparison of different methods.

While GALOIS demonstrates superior performance over baseline methods, I think the performance gain largely comes from the domain-specific knowledge encoded in atoms and program structures. Below are related questions.

1. How large is the program search space for each task? From Figure 1 in the appendix, I feel that the atom set is specialized for each task, though sometimes the atom set is shared among similar tasks. Then for each definite clause in code blocks, actually the head atom and the atom list in the body can also be fixed, and the model simply needs to predict whether there is a negation for each atom. In this case, the comparison to MPPS is a bit unfair, given that MPPS does not leverage a program sketch that already contains conditionals and loops.

2. In Section 4.2, I think the generalization capability largely comes from the design of subgoals, as shown in Figure 7. Related to my first question, I wonder how these subgoals are predicted in the first code block, because some of them are unnecessary if only considering the training environment.

3. How would GALOIS work on other benchmarks, e.g., those evaluated in the MPPS paper?

---

> ### Author Response · Authors · 2022-08-02
> **Response to Reviewer sTbQ (Part1/2)**
>
> **1. How large is the program search space for each task? From Figure 1 in the appendix, I feel that the atom set is specialized for each task, though sometimes the atom set is shared among similar tasks. Then for each definite clause in code blocks, actually the head atom and the atom list in the body can also be fixed, and the model simply needs to predict whether there is a negation for each atom. In this case, the comparison to MPPS is a bit unfair, given that MPPS does not leverage a program sketch that already contains conditionals and loops.**
>
> Regarding the search space, given a task of $n$ subgoal atoms, $m$ atoms of the Herbrand base (which are the states obtained from the environment, and we just simply convert them in the form of atoms), the size of its search space for $\text{??}\_\text{WHAT}$ would be $C_{m^2\cdot n}^{n}$; for $\text{??}\_\text{HOW}$, given $i$ atoms regarding numerical features, $j$ directional atoms, the size would be $C_{i^2\cdot j}^{j}$ (j=4 for the MiniGrid environment); for $\text{??}\_\text{WHAT}$, given $u$ descriptive body atoms and $v$ concrete action atoms, the size would be $C_{u^2\cdot v}^{v}$. Then the entire program search space size for each task would be $C_{m^2\cdot n}^{n}\cdot C_{i^2\cdot j}^{j}\cdot C_{u^2\cdot v}^{v}$ (e.g., for the UnlockPickup task, the size would be $40,758,502,400$ candidate rules in total).
>
> In practice, such an intractably large search space makes it impossible to predict the negation for each atom. On the other hand, we would like to clarify that, similar to GALOIS, MPPS also leverages a structure that already contains conditionals and loops implicitly (see $8^{th}$ paragraph in Section 2 in the MPPS paper [1]). Therefore, we believe the comparison is relatively fair.
> Even though MPPS is equipped with the conditional and loop constructs, it still performs less effective generalization ability than GALOIS (see experiment results in Table 1 \& 2 in the original paper, and newly added experimental results in table 2 below).
> This is because that MPPS's synthesizer is incapable of synthesizing programs with cause-effect logic (see line 42-57 in the original paper).
>
> **2. In Section 4.2, I think the generalization capability largely comes from the design of subgoals, as shown in Figure 7. Related to my first question, I wonder how these subgoals are predicted in the first code block because some of them are unnecessary if only considering the training environment.**
>
> We agree with the reviewer that the design of subgoals is important, however, we would like to clarify that an effective learning method for deriving effective cause-effect logic is also critical for good generalization. Note that, both GALOIS and MPPS use the same subgoals to synthesize programs, however, MPPS presents low generalizability as it fails to generate programs with cause-effect logic (see results in Table 1 and Table 2 in the original paper and newly added experiment results in table 2 below). This is because the strict imperative program synthesizer used by MPPS is incapable of synthesizing programs with cause-effect logic. In contrast, GALOIS includes both the imperative (program sketch) and declarative programming ($\partial$ILP) constructs, which is the key to the synthesis of generalizable cause-effect logic. Therefore, both the learning method and subgoals are critical for achieving good generalizability.
>
> Regarding the unnecessary subgoals, we assume that the reviewer refers to the `gt_goal` in the synthesized program in Figure 7 (right) in our paper while the UnlockPickUp environment does not have a `goal`. Sorry for the confusion caused, we need to clarify that the synthesized program in Figure 7 is the program learned for the BoxKey environment. And actually, the redundant (unnecessary) subgoals from the training environments would not affect since their corresponding rule would not be triggered according to the explicitly learned cause-effect relations goal.

---

> > ### Author Response · Authors · 2022-08-02
> > **Response to Reviewer sTbQ (Part2/2)**
> >
> > **3. How would GALOIS work on other benchmarks, e.g., those evaluated in the MPPS paper?**
> >
> > As suggested, to demonstrate the adaptability of our framework, we
> > further conduct experiments on the Craft-2D environment from previous
> > works [1, 2]. Specifically, we adopt
> > two original tasks: `get_gem` and `get_gold` in Craft-2D, because both
> > tasks require the agent to execute actions in a specific logic order in
> > order to accomplish the task. For `get_gem`, the agent needs to (1) get wood
> > and iron in the environment; (2) go to the workbench to produce a stick;
> > (3) go to the factory to make an axe with the stick and iron; (4) use
> > the axe to break a stone screen to get the gem. For `get_gold`, the agent
> > needs to (1) get wood and iron; (2) use the workbench to produce a plank
> > with the wood; (3) go to the toolshed to make a bridge; (4) cross a
> > river to get the gold. The results are shown in the following two
> > tables. We evaluate the GALOIS and MPPS in terms of the average return
> > and the number of steps required to finish the tasks. We report the mean
> > and standard deviation of the results over 5 random seeds. In Table 1,
> > we present the performance of the two methods on the original training
> > environment. Concretely, GALOIS and MPPS achieve similar performance,
> > which indicates that both methods can learn the comprehensive
> > task-solving logic. In Table 2, we evaluate the generalizability on test
> > environments with different task-solving logic ( semantic
> > modifications).  for `get_gem (sem-mod)`, the agent is initially
> > equipped with the stick and iron, thus it only needs to go to the
> > factory to make an axe and use it to get the gem directly instead of
> > collecting wood and iron from scratch. It is obvious that GALOIS
> > achieves much better generalizability in terms of average return and
> > steps due to its explicit use of *cause-effect* logic, whereas for the
> > MPPS model, since its execution is based on a fixed subprogram sequence,
> > the model performs poorly and fails to generalize. The results are
> > similar for the `get_gold` task.
> >
> > [*Table 1: Performance comparison*]
> >
> > |        |    get_gem   |              |   get_gold   |              |
> > |--------|:------------:|:------------:|:------------:|:------------:|
> > |        | GALOIS       | MPPS         | GALOIS       | MPPS         |
> > | Return | 0.418±0.025  | 0.408±0.020  | 0.410±0.030  | 0.404±0.035  |
> > | Step   | 60.988±2.709 | 64.704±2.055 | 62.284±3.338 | 64.992±3.120 |
> >
> > [*Table 2: Generalization comparison*]
> >
> > |        | get_gem (sem-mod) |              | get_gold (sem-mod) |              |
> > |--------|:-----------------:|:------------:|:------------------:|:------------:|
> > |        | GALOIS            | MPPS         | GALOIS             | MPPS         |
> > | Return | 0.806±0.024       | 0.327±0.012  | 0.792±0.022        | 0.397±0.019  |
> > | Step   | 21.300±2.411      | 72.480±1.105 | 22.792±2.203       | 66.272±2.089 |
> >
> > We hope that these comments have addressed the reviewer’s concerns about the paper. We are happy to answer any follow-up questions. We thank the reviewer's comments, this helps us a lot to improve the paper!
> >
> > [1] Yang Y, Inala J P, Bastani O, et al. Program synthesis guided reinforcement learning for partially observed environments[J]. Advances in Neural Information Processing Systems, 2021, 34: 29669-29683.
> >
> > [2] Andreas J, Klein D, Levine S. Modular multitask reinforcement learning with policy sketches[C]//International Conference on Machine Learning. PMLR, 2017: 166-175.

---

> ### Author Response · Authors · 2022-08-08
> **Resolving any pending concerns**
>
> We would really appreciate it if the reviewers can let us know whether they have any further concerns and whether our response addressed some/all of their concerns. We will try to address them before the discussion period ends. Thanks!

---

### Official Review · Reviewer_aGFV · 2022-07-07

**Rating:** 6
**Confidence:** 2
**Soundness:** 3 good
**Presentation:** 2 fair
**Contribution:** 2 fair

**Summary:**

This paper presents GALOIS, a novel method to synthetize interpretable sketch-based programs. The authors propose a new DSL combining features of declarative and imperative languages. This DSL enables learning to synthetize logic hierarchical programs which generalize better than their fully-neural counterparts. The authors also conducted an extensive evaluation of GALOIS on several MiniGrid environments showing better generalization capabilities, performance and reusability over multiple tasks.

**Questions:**

Summarizing some points expressed above:

- Which are the functions, namely the A set, available to both GALOIS and the baselines?
- How do you learn the sub-program needed by GALOIS (e.g., gt_key() or gt_door())? I assume these programs need to be learned somehow.
- Which is the DSL used by the neural baselines?
- Why is the proposed sketch design more robust or why is it able to generalize better over multiple problems? Basically, why did you choose to structure the sketch like that?
- How do you collect the train experiences for GALOIS? Do you run a partial sketch and do you keep only the successful trajectories?
- How is a GALOIS policy defined (see equation line 215)?


**Limitations:**

One potential drawback of this method is the fact that we are constrained to filling a program sketch. Therefore, I could imagine finding tasks for which the given sketch cannot be used successfully. Moreover, feature engineering is still required to define a good set of atoms (e.g., is_key(X)). Without a good set of atom features, then not even GALOIS would be able to solve certain tasks.

Lastly, learning causal-effect relationship in an environment is a problem which overlaps naturally with the topic of causality [1] in machine learning. In the past years there have been some initial works on causality-inspired RL agents (e.g., [2]), which can learn more profound causal-effect relationships by interacting with the environment. Although these works are tangential to program synthesis, it would be interesting to investigate how to learn program-based policies informed by a causality notion. In theory, we could be able to generate highly-generalizable programs without the need for ILP formalization and/or FOL formalization.

[1] Pearl, Judea. Causality. Cambridge university press, 2009.

[2] Dasgupta, I., Wang, J., Chiappa, S., Mitrovic, J., Ortega, P., Raposo, D., ... & Kurth-Nelson, Z. (2019). Causal reasoning from meta-reinforcement learning. arXiv preprint arXiv:1901.08162.


**Strengths And Weaknesses:**

The paper is interesting, since it enables us to generate highly-generalizable and explainable programs in an end-to-end fashion by providing also interpretable logical rules. As far as I know, the approach seems novel and I find it very impressive that we are able to synthesize a logical program directly instead of distilling it from a trained RL agent.

I think the paper is moving in the right direction, however, certain parts of the paper and some details are not very clear. For example, I found Section 3.2 and Section 3.3 a bit hard to follow. I also feel that the paper does not describe some initial assumptions (e.g., DSL used, how the baselines are trained) needed to understand the impact of GALOIS over DNN counterparts.

Program-sketches are heavily dependent on the problem we are trying to solve. Why is the proposed sketch more robust or why is it able to generalize better over multiple problems? I also mildly agree with the statement that “modifying or redesigning the sketch is also effortless.” (line 179-180). It does require quite a bit of effort and expert knowledge. If done wrong, it can hinder the overall final performance of the learned program.

On line 212, the paper says that the model is trained based on “the experience collected by interacting with the model”. How do you collect these experiences? Do you run a partial sketch and do you keep only the successful trajectories? How is a policy defined? Since it is a program that interacts with the environment, the actions are also deterministic, so there is no real policy.

I do not fully agree with the claim stated at lines 245-249. Performance depends heavily on the DSL used. For example, if the DSL does not provide any branching statement (e.g., if-then-else) based on the environment features, then clearly it is impossible to synthesize a correct program. GALOIS might just have a better and more descriptive DSL than its counterpart.  The paper and Appendix lack a description of the DSL used by the neural competitors, thus it is hard to understand if GALOIS is actually better in the MultiRoom environment.

Moreover, in the Appendix it is stated that the baselines use “2-layer multilayer perceptron (MLP) with 128 neurons in each layer” (lines 37-38). In my opinion, they hardly classify as deep neural networks. It would be interesting to see what happens if we increase the width/depth to more than 2 layers, thus having more realistic baselines. I would expect to see an increase in the convergence time, which would already justify why GALOIS is better, but also more realistic performance measurements. Additionally, regarding Section 4.3, worse performances on larger environments could also be caused by having shallow architectures on the baselines.

Minor issues:

Line 43: programs … have limited generalizability

Line 94: a procedure defined based…

Line 339: prorgam-synthesis —> program-synthesis

---

> ### Author Response · Authors · 2022-08-02
> **Response to Reviewer aGFV (Part1/4)**
>
> **1. Why is the proposed sketch design more robust or why is it able to generalize better over multiple problems? Why is the proposed sketch easy to be modified?**
>
> Many decision-making problems can be formulated as a hierarchical decision-making procedure (i.e., high-level and low-level controls) and solved via hierarchical decision-making algorithms like HRL [1,2,3]. Similarly, GALOIS follows the above principle and uses the program sketch to synthesize programs with two-level control loop(s). We believe GALOIS falls into the category of hierarchical decision-making algorithms, which are fit for solving a range of complex decision making problems [4,5].
>
> In fact, the proposed program-sketch only requires the synthesized program to follow the nested loops in the sketch while the detailed functions that should be done inside the loops are automatically learned. Such a program-sketch can be applied to many decision-making problems. Besides, to demonstrate the generalizability and adaptability of the proposed program sketch, we further conduct experiments on the Craft-2D environment using the same program sketch design, which also requires multi-step decision making and navigation (Please see the below block and table).
>
> >To demonstrate the adaptability of our framework, we further conduct experiments on the Craft-2D environment from previous works [6,7]. Specifically, we adopt two original tasks: `get_gem` and `get_gold` in Craft-2D, because both tasks require the agent to execute actions in a specific logic order in order to accomplish the task.
> For `get_gem`, the agent need to (1) get wood and iron in the environment; (2) go to the workbench to produce a stick; (3) go to the factory to make an axe with the stick and iron; (4) use the axe to break a stone screen to get the gem. For `get_gold`, agent need to (1) get wood and iron; (2) use the workbench to produce a plank with the wood; (3) go to the toolshed to make a bridge; (4) cross a river to get the gold. The results are shown in the following two tables. We evaluate the GALOIS and MPPS in terms of the average return and the number of steps required to finish the tasks. We report the mean and standard deviation of the results over 5 random seeds. In Table 1, we present the performance of the two methods on the original training environment. Concretely, GALOIS and MPPS achieve similar performance, which indicates that both methods can learn the comprehensive task-solving logic. In Table 2, we evaluate the generalizability on test environments with different task-solving logic (i.e., semantic modifications). E.g., for `get_gem (sem-mod)`, the agent is initially equipped with the stick and iron, thus it only needs to go to the factory to make an axe and use it to get the gem directly instead of collecting wood and iron from scratch. It is obvious that GALOIS achieves much better generalizability in terms of average return and steps due to its explicit use of cause-effect logic, whereas for the MPPS model, since its execution is based on a fixed subprogram sequence, the model performs poorly and fails to generalize. The results are similar for the `get_gold` task.
>
> [*Table 1*: Performance comparison]
>
> |        |    get_gem   |              |   get_gold   |              |
> |--------|:------------:|:------------:|:------------:|:------------:|
> |        | GALOIS       | MPPS         | GALOIS       | MPPS         |
> | Return | 0.418±0.025  | 0.408±0.020  | 0.410±0.030  | 0.404±0.035  |
> | Step   | 60.988±2.709 | 64.704±2.055 | 62.284±3.338 | 64.992±3.120 |
>
> [*Table 2*: Generalization comparison]
>
> |        | get_gem (sem-mod) |              | get_gold (sem-mod) |              |
> |--------|:-----------------:|:------------:|:------------------:|:------------:|
> |        | GALOIS            | MPPS         | GALOIS             | MPPS         |
> | Return | 0.806±0.024       | 0.327±0.012  | 0.792±0.022        | 0.397±0.019  |
> | Step   | 21.300±2.411      | 72.480±1.105 | 22.792±2.203       | 66.272±2.089 |
>
> Regarding the effort required to modify a sketch, designing this kind of structure is straightforward and scalable [8, 9]. Our sketch can be easily represented as a program with a nested while loop with any general-purpose programming languages (see Figure 7(right) in our paper). The sketch can remain unchanged for many tasks that require hierarchy to be solved. For certain tasks that hierarchy is not needed, like a simple navigation task, we can simply remove the outer loop of the sketch, which is rather straightforward and requires minor modifications.

---

> > ### Author Response · Authors · 2022-08-02
> > **Response to Reviewer aGFV (Part2/4)**
> >
> > **2. On line 212, the paper says that the model is trained based on “the experience collected by interacting with the model”. How is a policy defined? How do you collect these experiences? Do you run a partial sketch and do you keep only the successful trajectories?**
> >
> > The policy is defined as a stochastic policy expressed as a white-box program. it takes states as inputs and output action probability distribution. The only difference is that we map the input to a set of atoms using the encoder $E(\cdot)$ and use the decoder $D(\cdot)$ to map the output predicates to the original actions for the environment interaction. After the decoding, the action is sampled based on the output probability distribution (e.g., $a \sim D(p(E(s, \mathcal{L}_\mathrm{hybrid})))$, where $p(\cdot)$ denotes the deduction process and $\mathcal{L}_\mathrm{hybrid}$ is the proposed DSL). In this way, for each training iteration, GALOIS interacts with the environment to collect $n$ timesteps of data, including both successful and failed trajectories. Then the policy is updated via the Monte-Carlo policy gradient.
> >
> > **3. Which is the DSL used by the neural baselines? Performance depends heavily on the DSL used. GALOIS might just have a better and more descriptive DSL than its counterpart, thus it is hard to understand if GALOIS is actually better in the MultiRoom environment.**
> >
> > We agree with the reviewer that the performance depends on the DSL and being able to generate branching statements is critical to solving complex problems.
> > However, we would like to clarify that the DSL used by MPPS also contains branching/loop constructs implicitly (denoted as $\beta$ in the original paper). Yet its synthesizer only supports the synthesis of a fixed program sequence in which subprograms only come with post-condition (effect) but not pre-condition (cause) (refer to $8^{th}$ paragraph, Section 2 of MPPS paper [6]), i.e., successive subprograms would not be executed until previous subprograms are finished (see line 42-57). This is the reason why MPPS fails in the MultiRoom environment.
> > On the other hand, besides the sketch-based DSL that enables the branching statement, GALOIS also leverages a corresponding sketch-based ILP synthesizer that allows the synthesis of the generalizable program with explicit cause-effect logic (pre- and post-conditions), which is why GALOIS performs better in the MultiRoom Environment.

---

> > > ### Author Response · Authors · 2022-08-02
> > > **Response to Reviewer aGFV (Part3/4)**
> > >
> > > **4. Moreover, in the Appendix it is stated that the baselines use “2-layer multilayer perception (MLP) with 128 neurons in each layer”. It would be interesting to see what happens if we increase the width/depth to more than 2 layers, thus having more realistic baselines. Additionally, regarding Section 4.3, worse performances on larger environments could also be caused by having shallow architectures on the baselines.**
> > >
> > > We follow the choice of a ``2-layer'' multi-layer perception following previous works that achieved great success in multiple domains like games, and robot control [10,11]. We have included further experimental results on neural-network baselines (namely, DQN [12], SAC [10], and PPO [11]) with more layers on the UnlockPickup task. Specifically, we run experiments on MLP with 3 and 4 layers. The details are shown in the following tables. We report the mean and standard deviation of the results over 5 random seeds. As shown in the results (see below table), deepening network layers may improve PPO's performance on environments with varied sizes yet it is still no better than GALOIS, while the convergence time of them also increases (from approximately 2000 to 2500). However, in terms of generalizability, increasing network layers is counter-effective, we think one possible explanation is that the deeper neural network model could be over-parameterized and over-fitted to spurious bias which further hinders its generalizability.
> > >
> > > [*Table 3: Performance Comparison on UnlockPickUp Environment*]
> > > |           | DQN-3        | DQN-4        | SAC-3        | SAC-4        | PPO-3        | PPO-4        |
> > > | :-------: | :----------- | :----------- | :----------- | :----------- | :----------- | :----------- |
> > > | 6\*6 (tr) | 0\.185±0.054 | 0\.169±0.053 | 0\.544±0.111 | 0\.430±0.049 | 0\.842±0.045 | 0\.836±0.053 |
> > > | 8\*8      | 0\.223±0.014 | 0\.168±0.008 | 0\.506±0.121 | 0\.411±0.066 | 0\.908±0.007 | 0\.906±0.009 |
> > > | 10\*10    | 0\.181±0.044 | 0\.169±0.009 | 0\.483±0.088 | 0\.458±0.096 | 0\.932±0.015 | 0\.937±0.012 |
> > > | 12\*12    | 0\.164±0.027 | 0\.165±0.039 | 0\.494±0.107 | 0\.442±0.097 | 0\.948±0.006 | 0\.935±0.016 |
> > > | 14\*14    | 0\.205±0.032 | 0\.140±0.041 | 0\.467±0.149 | 0\.435±0.065 | 0\.960±0.005 | 0\.937±0.032 |
> > > | 16\*16    | 0\.161±0.030 | 0\.179±0.048 | 0\.485±0.102 | 0\.451±0.081 | 0\.960±0.005 | 0\.926±0.056 |
> > > | 18\*18    | 0\.177±0.049 | 0\.160±0.026 | 0\.498±0.104 | 0\.479±0.122 | 0\.958±0.012 | 0\.960±0.013 |
> > >
> > > [*Table 4: Generalization comparison on UnlockPickUp Environment*]
> > > |           | DQN-3        | DQN-4        | SAC-3        | SAC-4        | PPO-3        | PPO-4        |
> > > | :-------: | :----------- | :----------- | :----------- | :----------- | :----------- | :----------- |
> > > | 6\*6 (tr) | 0\.185±0.054 | 0\.169±0.053 | 0\.544±0.111 | 0\.430±0.049 | 0\.842±0.045 | 0\.836±0.053 |
> > > | sem-mod   | 0\.010±0.008 | 0\.011±0.007 | 0\.044±0.037 | 0\.061±0.058 | 0\.092±0.040 | 0\.082±0.025 |

---

> > > > ### Author Response · Authors · 2022-08-02
> > > > **Response to Reviewer aGFV (Part4/4)**
> > > >
> > > > **5. Which are the functions, namely the A set, available to both GALOIS and the baselines?**
> > > >
> > > > The A set of each corresponding hole functions consists of all candidate clauses, which are constructed by taking subgoals and actions as head atoms and Herbrand base representation of the state as body atoms. For example, assume that the predicate `pick()` in the hole function $??_\text{WHAT}$ can be deduced by the following three clauses (the A set for `pick()`):
> > > >
> > > > (1) `pick():- at(X) is_key(X), is_agent(Y), ¬has_key(Y)`;
> > > >
> > > > (2) `pick():- at(X) is_door(X), is_agent(Y), ¬has_key(Y)`;
> > > >
> > > > (3) `pick():- at(X) is_box(X), is_agent(Y), ¬has_key(Y)`.
> > > >
> > > > And for a whole environment, say UnlockPickUp, the functions are: `gt_key()`, `gt_door()`, `gt_box()`, `pick()`, `toggle()`, `drop()`, `left()`, `right()`, `up()`, `down()` and the A set contains all the candidate rules related to these functions.
> > > >
> > > > Note that the A set is actually unique for GALOIS, for a fair comparison, all baseline methods are provided with the same Herbrand base as input. For hierarchical baselines (hDQN [2], MPPS [6]), models are also provided with the same function set as output.
> > > >
> > > > **6. How do you learn the sub-program needed by GALOIS (e.g., gt_key() or gt_door())? I assume these programs need to be learned somehow.**
> > > >
> > > > The sub-programs are also learned using $\partial$ILP [13], which can automatically synthesize cause-effect programs. More precisely, each subprogram is associated with a set of candidate rules that are initialized with random weights. During each training iteration, the agent interacts with the environment by executing the candidate rules and sampling sub-programs from the corresponding output probability distribution (deduction). After collecting the experiences of $n$ times steps, the rule weights are updated with Monte-Carlo policy gradient during the training (induction) following the $\partial$ILP paradigm.
> > > >
> > > > We hope that these comments have addressed the reviewer’s concerns about the paper. We are happy to answer any follow-up questions. We thank the reviewer's comments, this helps us a lot to improve the paper!
> > > >
> > > > [1] Tessler C, Givony S, Zahavy T, et al. A deep hierarchical approach to lifelong learning in minecraft[C]//Proceedings of the AAAI Conference on Artificial Intelligence. 2017, 31(1).
> > > >
> > > > [2] Kulkarni T D, Narasimhan K, Saeedi A, et al. Hierarchical deep reinforcement learning: Integrating temporal abstraction and intrinsic motivation[J]. Advances in neural information processing systems, 2016, 29.
> > > >
> > > > [3] Andrychowicz M, Wolski F, Ray A, et al. Hindsight experience replay[J]. Advances in neural information processing systems, 2017, 30.
> > > >
> > > > [4] Nachum O, Gu S S, Lee H, et al. Data-efficient hierarchical reinforcement learning[J]. Advances in neural information processing systems, 2018, 31.
> > > >
> > > > [5] Eysenbach B, Gupta A, Ibarz J, et al. Diversity is all you need: Learning skills without a reward function[J]. arXiv preprint arXiv:1802.06070, 2018.
> > > >
> > > > [6] Yang Y, Inala J P, Bastani O, et al. Program synthesis guided reinforcement learning for partially observed environments[J]. Advances in Neural Information Processing Systems, 2021, 34: 29669-29683.
> > > >
> > > > [7] Andreas J, Klein D, Levine S. Modular multitask reinforcement learning with policy sketches[C]//International Conference on Machine Learning. PMLR, 2017: 166-175.
> > > >
> > > > [8] Solar-Lezama A. Program synthesis by sketching[M]. University of California, Berkeley, 2008.
> > > >
> > > > [9] Zhu H, Xiong Z, Magill S, et al. An inductive synthesis framework for verifiable reinforcement learning[C]//Proceedings of the 40th ACM SIGPLAN conference on programming language design and implementation. 2019: 686-701.
> > > >
> > > > [10] Haarnoja T, Zhou A, Abbeel P, et al. Soft actor-critic: Off-policy maximum entropy deep reinforcement learning with a stochastic actor[C]//International conference on machine learning. PMLR, 2018: 1861-1870.
> > > >
> > > > [11] Schulman J, Wolski F, Dhariwal P, et al. Proximal policy optimization algorithms[J]. arXiv preprint arXiv:1707.06347, 2017.
> > > >
> > > > [12] Mnih V, Kavukcuoglu K, Silver D, et al. Playing atari with deep reinforcement learning[J]. arXiv preprint arXiv:1312.5602, 2013.
> > > >
> > > > [13] Evans R, Grefenstette E. Learning explanatory rules from noisy data[J]. Journal of Artificial Intelligence Research, 2018, 61: 1-64.

---

> ### Author Response · Authors · 2022-08-08
> **Resolving any pending concerns**
>
> We would really appreciate it if the reviewers can let us know whether they have any further concerns and whether our response addressed some/all of their concerns. We will try to address them before the discussion period ends. Thanks!

---

> > ### Comment · Reviewer_aGFV · 2022-08-08
> > **Answer to the author response**
> >
> > I would like to thank the authors for the extensive rebuttal answer that addressed some of my concerns. I raised my score to a weak accept.
> >
> > Some food for thought to engage a little more in the discussion.
> >
> > I have a general question regarding employing a program sketch. In general, I think it provides a very powerful inductive bias that can simplify the problem quite a lot. Therefore, the generalization and better performances might be caused mostly by having a well-designed program sketch, rather than a good program synthesis algorithm. Since the program sketch is designed by humans, I wonder if this approach might be applicable to more complex problems, where designing the program sketch is hard. Lastly, it would be nice to understand if it would be possible to design a two-step process, in which we learn a program sketch and then we fill it with the correct actions.

---

> > > ### Author Response · Authors · 2022-08-09
> > > **Response to the concern**
> > >
> > > We thank the reviewer for the recognition of our work!
> > >
> > > This work aims at synthesizing a non-strict declarative program (to our knowledge) for the first time. For this, the program sketch-based synthesis paradigm is required as it is one of the most effective ways to generate complex programs (with control flow) [1]. In practice, we cannot synthesize an non-strict declarative program without a program sketch.
> > >
> > > Besides, we agree with the reviewer that, in extremely complex problems, a non-well-designed program sketch may affect the performance. This could be a potential limitation of our method, especially when the program sketch is not given as prior. As mentioned by the reviewer, we would like to study how to automatically generate and dynamically optimize a program sketch for future work.
> > >
> > > In our opinion, the idea of an end-to-end two-step framework that automatically learns and dynamically optimizes an imperative program sketch and the corresponding declarative hole functions for the decision-making tasks is possible and an extremely promising direction. We would love to investigate it in our future work.
> > >
> > > We hope that these comments have addressed the reviewer's concern. We are happy to answer any follow-up concerns. We thank the reviewer's comments, this helps us a lot to improve the paper!
> > >
> > > [1] Solar-Lezama A. Program synthesis by sketching[M]. University of California, Berkeley, 2008.

---

### Official Review · Reviewer_WqvU · 2022-07-10

**Rating:** 6
**Confidence:** 3
**Soundness:** 3 good
**Presentation:** 2 fair
**Contribution:** 3 good

**Summary:**

GALOIS introduces a DSL synthesis framework for reinforcement learning that combines both imperative and declarative programs.
It takes in a sketch of proof, in the form of an inter-procedural control flow graph, then fills in the "holes" in the graph via synthesizing declarative programs.
The contributions of this paper include:
1. A new domain-specific language for synthesizing a hierarchical program
2. A synthesize algorithm of this DSL
3. Evaluations comparing GALOIS and various baselines, showing GALOIS has better performance.

**Questions:**

1. Can you explain more about the predicate deduction process, and the $normalized weighted vector $\theta_\rho^\psi$?
2. Can you explain the user interface on the framework. The user requires to provide the ICFG, but what about the head nodes?
3. In general, how is the action space incorporated into the program synthesis process?
4. It would be helpful to include another environment to demonstrate adaptability as a general framework.

**Limitations:**

The authors addressed the limitations of this approach.

**Strengths And Weaknesses:**

Originality: 3/5

The paper designed a hybrid DSL for the reinforcement learning environment. Although template-based reinforcement learning and template generation are studied under a reinforcement learning setup, this work focuses on causal-effect by combining the imperative program and declarative program together, which is not studied before.

Quality: 3/5

Pros:
1. This paper shows a variety of baseline comparisons on the box task.
Cons:
1. The environment only has the "box-world" environment evaluation, but not another one.
     a) What are the user input and difficulties to adapt this approach to another setup?
     b) Performance comparison (minor)

Clarity: 2.5/5

Pros: The intuition and $DSL_{hybrid}$ are explained pretty clearly.
Cons:
1. The formalization is not clear enough. Specifically,
    a) How is the action space incorporate into the program synthesis is not clear in the formalization.
    b) How the rule head atoms are selected is not clear, such as "gt_key". What's more, how are they semantically aligned with the rule body?
    c) Can you explain more about the "normalized weighted vector $\theta_\rho^\psi$?
    d) It would help a lot to include a running example

Significance: 3/5
This approach performs much better than both the pure neural approach and the current program synthesis approach.

---

> ### Author Response · Authors · 2022-08-02
> **Response to Reviewer WqvU (Part1/2)**
>
> We thank the reviewer for the insightful and useful feedback, please see the following for our response.
>
> **1. Can you explain more about the normalized weighted vector
> $\theta_\rho^\psi$ and the predicate deduction process?**
>
> The normalized weight vector $\theta_\rho^\psi$ is a vector for the
> intensional predicate $\psi$ in the hole function $\rho$, which consists
> of weights for all its corresponding candidate clauses (i.e., rules).
> Intuitively, the weight denotes whether the corresponding candidate
> clause is the most appropriate rule for $\psi$. For example, assume that
> the predicate `pick()` in the hole function $\text{??}\_\text{WHAT}$ has three
> candidate clauses:\
> (1) `pick():- at(X), is_key(X), is_agent(Y), ¬has_key(Y)`;\
> (2) `pick():- at(X), is_door(X), is_agent(Y), ¬has_key(Y)`;\
> (3) `pick():- at(X), is_box(X), is_agent(Y), ¬has_key(Y)`.\
> The weight vector for predicate `pick` is
> $\theta_\text{WHAT}^{pick} = [\theta_\text{WHAT}^{pick, 1}, \theta_\text{WHAT}^{pick, 2}, \theta_\text{WHAT}^{pick, 3}]$,
> where each weight $\theta_\text{WHAT}^{pick, i}, i\in[1,2,3]$ measures the
> degree of the clause $i$ being the most appropriate one. Evidently,
> clause (1) is the most appropriate one as the key can only be picked
> when `is_key(X)` is true. Thus, to automatically discover the
> most appropriate clause(s) (e.g., searching a
> $\theta_\text{WHAT}^{pick}$ where the $\theta_\text{WHAT}^{pick, 1}$ is
> higher than the others), we adopt the predicate deduction process.
>
> We leverage the $\partial$ILP [1] paradigm which
> implements this process via gradient update. Concretely, during training
> iteration, the predicate deduction step in $\partial$ILP is applied to
> the body of the clauses ( `at(X)`, `is_key(X)`, `is_agent(Y)`,
> `¬has_key(Y)`) to derive the weight of each candidate clause. Then the
> predicate is sampled based on the weights (normalized as a probability
> distribution) and the corresponding action will be performed in the
> environment to collect experiences. Finally, based on the experiences
> collected, the weight vector $\theta_\rho^\psi$ is updated via
> back-propagation.
>
> As a result, in the `pick()` example, during the training, the weight of
> the right rule ($\theta_\text{WHAT}^{pick, 1}$) will be increased and
> the wrong rules ($\theta_\text{WHAT}^{pick, 2}$,
> $\theta_\text{WHAT}^{pick, 3}$) will be gradually decreased. The correct
> clause (clause (1)) will finally be learned.
>
> **2. Can you explain the user interface on the framework. The user
> requires to provide the ICFG, but what about the head nodes?**
>
> For user interface, the general program sketch (in the form of ICFG
> shown in Figure 4) and part of the head nodes (i.e., predicates) are
> provided by human. The head nodes (i.e., predicates) consist of two
> types: subgoals ( `gt_key()`) and original actions in the environment
> (`left()`, `right()`, `up()`, `down()`, `pick()`). The actions in the
> environment are automatically mapped to predicate form. Only the
> subgoals are required to be provided by human, which is a standard
> approach as a fully end-to-end framework that automatically learns the
> subgoals is hard and remains unsolved [2].
>
> **3. In general, how is the action space incorporated into the program
> synthesis process?**
>
> The action space is mapped to be the head nodes (predicates) of the
> synthesized clauses. For example, if the original action space of the
> environment is \[left, right, up, down\], we map them into 4 head
> nodes: \[`left()`, `right()`, `up()`, `down()`\]. These head nodes are
> associated with all possible candidate clauses initialized with random
> weights ( refer to the previously mentioned `pick()` example). During
> the program synthesis process, the weight of the correct rules will
> increase and the weights of the wrong rules will decrease, leading to a
> final program with correct cause-effect logic (expressed as rules).

---

> > ### Author Response · Authors · 2022-08-02
> > **Response to Reviewer WqvU (Part2/2)**
> >
> > **4. It would be helpful to include another environment to demonstrate
> > adaptability as a general framework.**
> >
> > As suggested, to demonstrate the adaptability of our framework, we
> > further conduct experiments on the Craft-2D environment from previous
> > works [2, 3]. Specifically, we adopt
> > two original tasks: `get_gem` and `get_gold` in Craft-2D, because both
> > tasks require the agent to execute actions in a specific logic order in
> > order to accomplish the task. For `get_gem`, agent need to (1) get wood
> > and iron in the environment; (2) go to the workbench to produce a stick;
> > (3) go to the factory to make an axe with the stick and iron; (4) use
> > the axe to break a stone screen to get the gem. For `get_gold`, agent
> > need to (1) get wood and iron; (2) use the workbench to produce a plank
> > with the wood; (3) go to the toolshed to make a bridge; (4) cross a
> > river to get the gold. The results are shown in the following two
> > tables. We evaluate the GALOIS and MPPS in terms of the average return
> > and the number of steps required to finish the tasks. We report the mean
> > and standard deviation of the results over 5 random seeds. In Table 1,
> > we present the performance of the two methods on the original training
> > environment. Concretely, GALOIS and MPPS achieves similar performance,
> > which indicates that both methods can learn the comprehensive
> > task-solving logic. In Table 2, we evaluate the generalizability on test
> > environments with different task-solving logic ( semantic
> > modifications).  for `get_gem (sem-mod)`, the agent is initially
> > equipped with the stick and iron, thus it only needs to go to the the
> > factory to make an axe and use it to get the gem directly instead of
> > collecting wood and iron from scratch. It is obvious that GALOIS
> > achieves much better generalizability in terms of average return and
> > steps due to its explicit use of *cause-effect* logic, whereas for the
> > MPPS model, since its execution is based on a fixed subprogram sequence,
> > the model performs poorly and fails to generalize. The results are
> > similar for the `get_gold` task.
> >
> > [*Table 1: Performance comparison*]
> > |        |    get_gem   |              |   get_gold   |              |
> > |--------|:------------:|:------------:|:------------:|:------------:|
> > |        | GALOIS       | MPPS         | GALOIS       | MPPS         |
> > | Return | 0.418±0.025  | 0.408±0.020  | 0.410±0.030  | 0.404±0.035  |
> > | Step   | 60.988±2.709 | 64.704±2.055 | 62.284±3.338 | 64.992±3.120 |
> >
> > [*Table 2: Generalization comparison*]
> > |        | get_gem (sem-mod) |              | get_gold (sem-mod) |              |
> > |--------|:-----------------:|:------------:|:------------------:|:------------:|
> > |        | GALOIS            | MPPS         | GALOIS             | MPPS         |
> > | Return | 0.806±0.024       | 0.327±0.012  | 0.792±0.022        | 0.397±0.019  |
> > | Step   | 21.300±2.411      | 72.480±1.105 | 22.792±2.203       | 66.272±2.089 |
> >
> > [1] Evans, Richard, and Edward Grefenstette. "Learning explanatory rules from noisy data." Journal of Artificial Intelligence Research 61 (2018): 1-64.
> >
> > [2] Yang, Yichen, et al. "Program synthesis guided reinforcement learning for partially observed environments." Advances in Neural Information Processing Systems 34 (2021): 29669-29683.
> >
> > [3] Andreas, Jacob, Dan Klein, and Sergey Levine. "Modular multitask reinforcement learning with policy sketches." International Conference on Machine Learning. PMLR, 2017.
> >
> > We hope that these comments have addressed the reviewer’s concerns about the paper. We are happy to answer any follow-up questions. We thank the reviewer's comments, this helps us a lot to improve the paper!

---

> > ### Comment · Reviewer_WqvU · 2022-08-08
> > **A bit more concerns**
> >
> > You have addressed my concerns and I have raised my rating to weak accept.
> >
> > There is a bit more concern here. As you have addressed in response to Reviewer sTbQ, there is a large search space for the program synthesis procedure. How does that affect the learning efficiency in terms of time consumption?

---

> > > ### Author Response · Authors · 2022-08-09
> > > **Response to the concern**
> > >
> > > We thank the reviewer for the recognition of our work!
> > >
> > > Intuitively, for almost all program synthesis algorithms, a larger search space generally requires more computational time; we, however, find that GALOIS and all baselines in Fig.6 take similar time consumption to finish the policy learning within 5000 episodes (i.e., about 0.5 hours for DoorKey, 1 hour for BoxKey, UnlockPickup, MultiRoom, respectively). This supplement finding reveals that, compared to related baselines, GALOIS achieves better performance and learning efficacy while causing no significant computational burden.
> > >
> > > We hope that these comments have addressed the reviewer’s concern. We are happy to answer any follow-up concerns. We thank the reviewer's comments, this helps us a lot to improve the paper!

---

> ### Author Response · Authors · 2022-08-08
> **Resolving any pending concerns**
>
> We would really appreciate it if the reviewers can let us know whether they have any further concerns and whether our response addressed some/all of their concerns. We will try to address them before the discussion period ends. Thanks!

---

### Meta-Review · Area_Chair_sXrR · 2022-08-25

**Recommendation:** Accept
**Confidence:** Less certain

**Metareview:**

All reviewers liked the presented GALOIS framework to synthesize hierarchical and cause-effect logic programs that represent policy for solving a task. The use of program sketches to represent the search space and using policy gradients to learn the programs is also quite interesting. However, there were some concerns about the generality of the approach, the amount of human knowledge required in the provided sketch, limited evaluation environments, and scalability of the synthesis approach. The author response helped quite a bit towards many of these concerns. It would be great to incorporate the additional discussions and evaluations from the response in the next paper version.

**Award:**

No

---

### Decision · Program_Chairs · 2022-09-14

Accept